# TOPOVIBL-REC114 interaction regulates meiotic DNA double-strand breaks

Alexandre Nore [1,5], Ariadna B. Juarez-Martinez [2,5], Julie Clément [1], Christine Brun[1], Boubou Diagouraga[3], Hamida Laroussi[2], Corinne Grey[1], Henri Marc Bourbon[4], Jan Kadlec [2] ✉, Thomas Robert [3] ✉ & Bernard de Massy [1] ✉

Meiosis requires the formation of programmed DNA double strand breaks (DSBs), essential for fertility and for generating genetic diversity. DSBs are induced by the catalytic activity of the TOPOVIL complex formed by SPO11 and TOPOVIBL. To ensure genomic integrity, DNA cleavage activity is tightly regulated, and several accessory factors (REC114, MEI4, IHO1, and MEI1) are needed for DSB formation in mice. How and when these proteins act is not understood. Here, we show that REC114 is a direct partner of TOPOVIBL, and identify their conserved interacting domains by structural analysis. We then analyse the role of this interaction by monitoring meiotic DSBs in female and male mice carrying point mutations in TOPOVIBL that decrease or disrupt its binding to REC114. In these mutants, DSB activity is strongly reduced genome-wide in oocytes, and only in sub-telomeric regions in spermatocytes. In addition, in mutant spermatocytes, DSB activity is delayed in autosomes. These results suggest that REC114 is a key member of the TOPOVIL catalytic complex, and that the REC114/TOPOVIBL interaction ensures the efficiency and timing of DSB activity.

Sexual reproduction relies on the specialized cell division of meiosis to generate haploid cells that eventually differentiate into gametes. In most taxa, proper segregation of the homologous chromosomes (homologs) depends on homologous recombination, which physically connects homologs through at least one reciprocal exchange (crossover) between each homolog pair[1]. Meiotic recombination initiates at the onset of prophase I by the formation of hundreds of programmed DNA double-strand breaks (DSBs) at preferred DNA sites, named hotspots[2]. DSBs are formed by the collective action of a conserved set of proteins: the TOPOVIL complex and its accessory partners[3–6].

The TOPOVIL complex is evolutionarily related to the TopoVI type IIB topoisomerase[7], and is composed of two conserved subunits: the SPO11 catalytic subunit and TOPOVIBL. TOPOVIBL shares partial homology with the GHKL-ATPase domain (thus named GHKL-like), with the central transducer domain and to a lesser extent, with the regulatory C-terminal domain (CTD) of the TopoVIB subunit of archaeal TopoVI[5,6,8–11]. The TOPOVIL meiosis-specific activity is finely regulated in time (i.e., must be turned on and off at precise time windows) and in space (i.e., active at specific chromosomal locations) during meiotic prophase. However, the molecular mechanisms underlying this complex regulation remain largely elusive.

In *Saccharomyces cerevisiae*, Rec102, the identified TOPOVIBL homolog, shares only partial similarity with TopoVIB, with the conserved transducer domain but not with the GHKL-like domain[5]. Interaction and biochemical studies suggest that Rec104, which is also essential for DSB formation in yeast, could replace the GHKL domain and that the Rec102/Rec104 complex fulfils the function of TOPOVIBL[12–14]. The *S. cerevisiae* core complex defined biochemically is

[1]Institut de Génétique Humaine (IGH), Centre National de la Recherche Scientifique, Univ Montpellier, Montpellier, France. [2]Univ. Grenoble Alpes, CNRS, CEA, IBS, F-38000 Grenoble, France. [3]CBS, Univ Montpellier, CNRS, INSERM, Montpellier, France. [4]Centre de Biologie Intégrative, CNRS, Université de Toulouse, Toulouse, France. [5]These authors contributed equally: Alexandre Nore, Ariadna B. Juarez-Martinez. ✉e-mail: jan.kadlec@ibs.fr; thomas.robert@cbs.cnrs.fr; bernard.de-massy@igh.cnrs.fr

composed of Spo11, Rec102, Rec104 and Ski8, a protein directly interacting with Spo11[13]. In addition, to the core complex, several accessory proteins are required for DSB formation. In *S. cerevisiae*, these accessory partners are Rec114, Mei4 and Mer2, which interact with DNA that mediates the formation of condensate-like RMM protein clusters[15]. These condensates might recruit the Spo11/Rec102/Rec104/Ski8 complex to DNA through an interaction of Rec114 with Rec102/Rec104[12,15,16]. The finding that the Rec114 residues involved in this interaction are essential for DSB formation supports this hypothesis[15].

In *M. musculus*, SPO11 and TOPOVIBL are evolutionarily conserved[5,17,18] and the DSB sites are determined by PRDM9 that recognizes specific DNA motifs and modifies chromatin upon binding to these sites[19–21]. In the mouse, the accessory proteins REC114, MEI4, IHO1 (orthologues of *S. cerevisiae* Rec114, Mei4 and Mer2, respectively), and MEI1 are essential for DSB formation, and localize as foci on chromosome axes at meiotic prophase onset. It has been proposed that IHO1, MEI4 and REC114 directly control TOPOVIL through its recruitment or activation[22–26]. MEI4 and REC114 form a stable complex, and structural analyses revealed that REC114 N-terminus forms a Pleckstrin homology (PH) domain[25,27] harbouring exposed conserved residues potentially involved in protein-protein interactions. This suggests that REC114 acts as a regulatory platform. In line with this hypothesis, it was recently reported that ANKRD31 directly interacts with REC114 PH domain and is involved in regulating DSB number and localization[27,28]. However, how these accessory proteins participate in TOPOVIL activity remains to be determined.

Here, using structural analysis, we found that the mouse REC114 PH domain directly interacts with a conserved C-terminal peptide of TOPOVIBL, identifying its CTD as a predicted regulatory unit of the TOPOVIL complex. Accordingly, in mice where TOPOVIBL interaction with REC114 was disrupted, we observed meiotic DSB formation defects in both sexes associated with reduced fertility. In females, DSB formation was drastically reduced. In males, DSB formation was delayed, but DSB levels were not reduced except for sub-telomeric regions. On the X and Y chromosomes, which recombine specifically in the distal sub-telomeric pseudo-autosomal region (PAR) region, DSB activity reduction led to chromosome synapsis defects. These results suggest that REC114 is part of the TOPOVIL catalytic complex and acts to regulate the level and timing of DSB formation.

## Results

### TOPOVIBL forms a stable complex with REC114

Using yeast two-hybrid assays (Y2H) we identified a specific interaction between mouse TOPOVIBL and REC114. The deletion analysis showed that the N-terminal PH domain of REC114 was required for this interaction because TOPOVIBL did not bind to REC114 lacking the first 39 amino acids (40–259) (Fig. 1a, b and Supplementary Fig. 1a). This interaction also required TOPOVIBL C-terminus as indicated by the absence of binding upon deletion of its last 29 amino acids (TOPOVIBL[1–550]) (Fig. 1a, c and Supplementary Fig. 1b). TOPOVIBL[1–550] could still interact with SPO11β, indicating that the deletion of the 29 amino acids did not disrupt TOPOVIBL folding (Fig. 1c and Supplementary Fig. 1b). We obtained similar results in vitro using Strep-tag pull-down assays (Fig. 1d). While it was difficult to produce full-length TOPOVIBL, we could express and purify a 6xHis-SUMO fusion of TOPOVIBL that lacks part of its transducer domain (construct 1–385; Fig. 1d, lane 2) and also the 6xHis-TOPOVIBL C-terminal domain (CTD; 452–579) (Fig. 1d, lane 3). REC114 clearly interacted with TOPOVIBL[452–579] (Fig. 1d, lane 12), but not with TOPOVIBL[1–385] (Fig. 1d, lane 11). The REC114 N-terminal domain (REC114[1–159]), which includes the PH domain, was sufficient for the interaction with TOPOVIBL[452–579] (Fig. 1d, lane 14). In agreement, the REC114 PH domain and TOPOVIBL[452–579] co-eluted in a single peak during size exclusion chromatography (Fig. 1e and Supplementary Fig. 2a).

The structure prediction analysis and AlphaFold2 model (AF-J3QMY9) suggested that TOPOVIBL CTD does not contain any known globular domain and was partially intrinsically disordered. However, it indicated the presence of a putative helix at its C-terminus. Interestingly, within the last C-terminal residues, we detected a high conservation among metazoan species (Fig. 1f). An extensive analysis of TOPOVIBL phylogeny indeed identified that the three helices of the transducer domain and this predicted C-terminal helix are the main conserved regions of TOPOVIBL in metazoans[7]. As our yeast two-hybrid assays suggested that the corresponding C-terminal 29 residues of TOPOVIBL (residues 551 to 579) were important for the interaction with REC114 (Fig. 1c), we hypothesized that this predicted conserved helix (residues 559–572, Fig. 1f) might represent the REC114 binding region. Using isothermal titration calorimetry (ITC), we showed that REC114 PH domain bound to TOPOVIBL CTD (452–579) with a dissociation constant ($K_d$) of 1.2 μM (Fig. 1g). Moreover, the interaction between a TOPOVIBL peptide spanning residues 559–576 and REC114 PH domain was in the same range ($K_d$ = 3.3 μM) (Fig. 1h). These results demonstrate that this highly conserved motif in TOPOVIBL CTD (559–576) is sufficient for interaction with REC114. Whether residues in the 452–559 region of TOPOVIBL contribute to the interaction with REC114 cannot be excluded, but no conserved motif could be identified in this interval (Supplementary Fig. 3). This question is also addressed by the mutational analysis presented below.

### Crystal structure of the TOPOVIBL-REC114 complex

We then determined the crystal structure of the complex formed by the REC114 PH domain (residues 15–159) and the TOPOVIBL[559–576] peptide by X-ray crystallography (2.3 Å resolution, $R_{free}$ of 24.9%, and R-factor of 22.6%) (Supplementary Table 1, Supplementary Fig. 2b). The structure of the TOPOVIBL-bound REC114 PH domain (two perpendicular antiparallel β-sheets followed by a C-terminal helix) was essentially the same as in its unbound form[25]. The TOPOVIBL C-terminal peptide folds into a single helix that interacts with the REC114 PH domain β-sheet formed of strands β1, β2 and β6-β8, burying 731 Å² of surface area (Fig. 2a, b). The interaction surface on REC114 is formed of highly conserved surface residues (Fig. 2c and Supplementary Fig. 2c). In the N-terminal part of the TOPOVIBL peptide, L561, W562 and V566 pack against a hydrophobic surface of REC114 (Fig. 2d). Specifically, W562 is located in a hydrophobic pocket formed by aliphatic side chains of K95, V97, L104 and M115 of REC114. In the central part of the TOPOVIBL helix, L569 binds to a hydrophobic groove in REC114 formed by V97, R99, C102, L104, and R117 (Fig. 2e). The well-conserved R99 and R117 residues form several hydrogen bonds with main-chain carbonyls in the TOPOVIBL helix, and R117 forms a salt bridge with E571. The C-terminal part of the TOPOVIBL peptide forms a 3₁₀ helix where W572 inserts into another hydrophobic pocket of REC114 made of R24, V53, C102 and R117 (Fig. 2f). W572 also forms cation-π interactions with the guanidinium groups of the two arginine residues and main-chain hydrogen bonds with R24 and Q119. Finally, L573 packs against a hydrophobic surface around M100 (Fig. 2f).

Most of the TOPOVIBL residues involved in the interaction with REC114 are highly conserved across metazoan species (Fig. 1f). We showed the interaction conservation by modelling the complex from different vertebrate species using AlphaFold2[29] (Supplementary Fig. 4a). To identify the TOPOVIBL residues required for the interaction with REC114, we mutated several candidate residues (W562, V566, L569 and W572) based on structure and conservation. In pull-down assays with the Strep-tagged REC114 PH domain, mutations W562A, W562E, V566R, L569R and W572L led to undetectable interaction with TOPOVIBL (Fig. 2g, lanes 9–13). We also mutated two hydrophobic residues of the REC114 β-sheet (V97D, L104D). Both mutations lead to undetectable interaction with TOPOVIBL (Supplementary Fig. 4b, lanes 9 and 10). The mutations introduced in TOPOVIBL and REC114 did not alter the structure of the proteins as judged by gel filtration

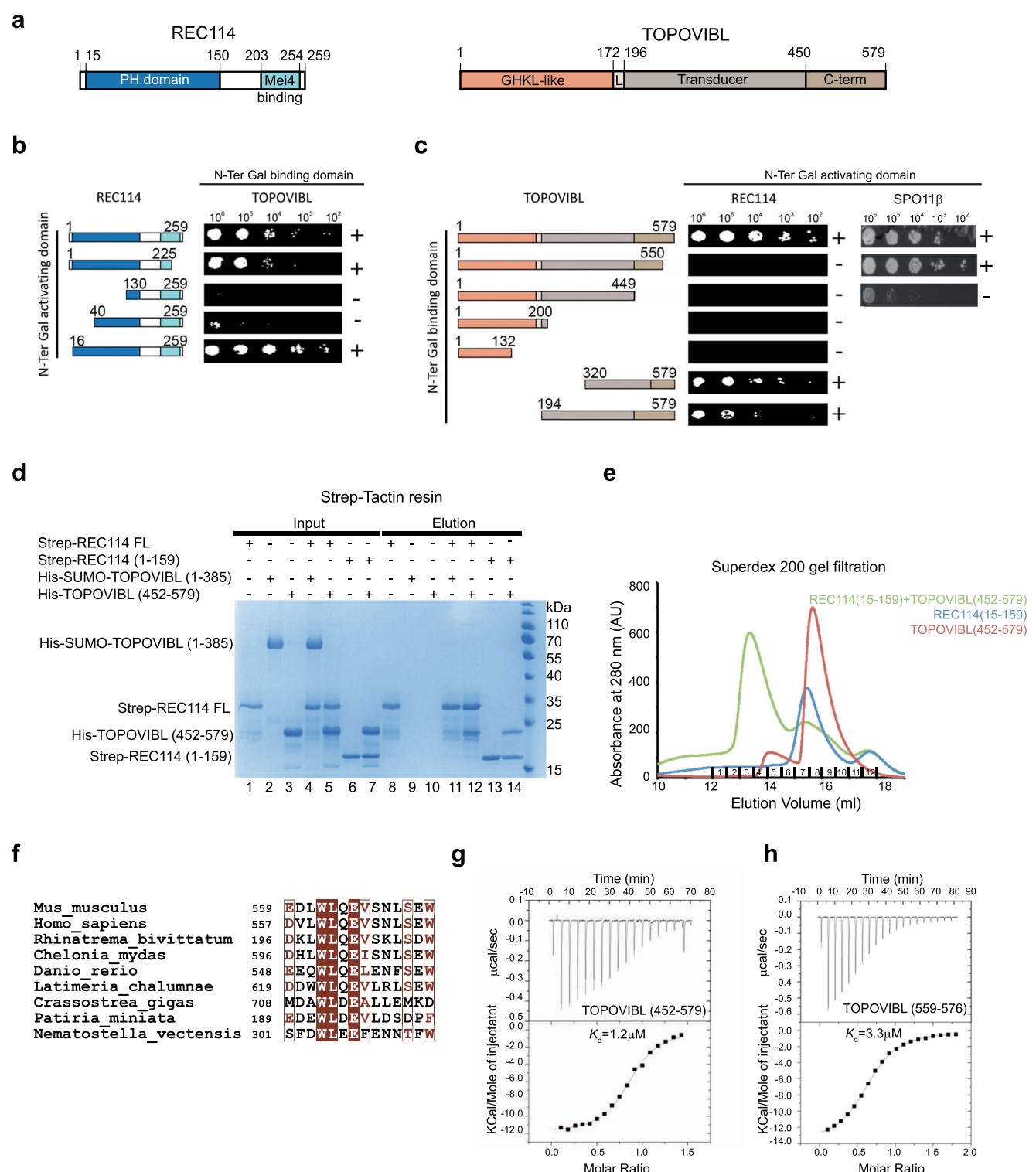

analysis (Supplementary Figs. 4c, d, 5a, b). We confirmed the role of TOPOVIBL W562 in the interaction with REC114 by ITC measurement and showed that both the TOPOVIBL[559–576] peptide and the TOPOVIBL[452–579] purified protein containing the W562A mutation did not bind to REC114 PH domain (Supplementary Fig. 4e, f). This indicates an essential role for W562 in the interaction with the REC114 PH domain and the absence of any significant interaction in the adjacent 452–559 region of TOPOVIBL. Moreover, in yeast two-hybrid assays using full-length proteins we showed that W562A and W562G led to undetectable interaction between TOPOVIBL and REC114 (Fig. 2h and Supplementary Fig. 1c). We conclude that the TOPOVIBL W562A

mutation specifically disrupts the interaction with REC114 in vitro, and we selected this mutation for the in vivo studies (see below).

## REC114 partners

Mouse REC114 directly interacts also with MEI4[25] and ANKRD31[27]. The REC114/MEI4 interaction does not involve the REC114 PH domain but the C-terminal domain of REC114 (203–254)[25]. In agreement, we could show by size exclusion chromatography the simultaneous interaction of REC114 with MEI4 and TOPOVIBL (Fig. 2i and Supplementary Fig. 5c). Conversely, the REC114/ANKRD31 interaction involves the REC114 PH domain. The crystal structure of the ANKRD31-REC114 complex[27]

**Fig. 1 | The C-terminal region of TOPOVIBL interacts with the N-terminal domain of REC114. a** Schematic representation of the domain structure of mouse REC114 (left) and TOPOVIBL (right). PH: Pleckstrin homology domain, MEI4 binding region, GHKL-like: Gyrase, HSP90, Histidine Kinase, MutL domain, L: linker, C-term: C-terminal. **b** Yeast two-hybrid assays showed that REC114 N-terminal domain is required for the interaction with TOPOVIBL. Full length and truncated REC114 proteins were tested for interaction with full length TOPOVIBL. Growth (+ or −) was assayed on medium lacking leucine, tryptophane and histidine and with 5 mM 3-amino-1,2,4-triazole. **c** Yeast two-hybrid assays indicated that TOPOVIBL last 29 residues are required for the interaction with REC114 but not with SPO11β. Full length and truncated TOPOVIBL proteins were tested for interaction with full length REC114 and SPO11β. Growth (+ or −) was assayed on medium lacking leucine, tryptophane and histidine. **d** Pull-down experiments of Strep-tagged REC114 with TOPOVIBL domains. All proteins were first purified by affinity chromatography and

gel filtration. Proteins were mixed as indicated above the lanes. A total of 0.8% of the input (lanes 1−7) and 1.2% of the eluates (lanes 8−14) were analyzed on 12% SDS-PAGE gels stained with coomassie brilliant blue. TOPOVIBL (1-385) is not retained by REC114 (compare lanes 4 and 11), whereas TOPOVIBL C-terminal region (452-579) is sufficient for the interaction with FL-REC114 (compare lanes 5 and 12) as well as REC114[1−159] (compare lanes 7 and 14). Source data are provided as a Source Data file. **e** Overlay of Superdex 200 gel filtration elution profiles of REC114[15−159], TOPOVIBL[452−579], and their complex. SDS-PAGE gels with eluted fractions are shown in Supplementary Fig. 2a. **f** Sequence alignment of the 14-aa conserved motif at the TOPOVIBL C-terminus in metazoans. Brown letters: equivalent amino acids; white letters: identical amino acids. **g** ITC measurement of the interaction affinity between REC114[15−159] and TOPOVIBL[452−579]. **h** ITC measurement of the interaction affinity between REC114[15−159] and TOPOVIBL[559−576].

shows that the ANKRD31 interacting fragment (45 residues) is significantly longer than that of TOPOVIBL and covers a larger surface on REC114 (~1800 Å²) packing against both of its β-sheets (Supplementary Fig. 5d, e). The C-terminal part of the ANKRD31 peptide forms two helices and packs against the same surface as TOPOVIBL, and both peptides interact with equivalent REC114 residues (Fig. 2j). This structural comparison suggests mutually exclusive binding of these two proteins to REC114, likely with higher affinity for ANKRD31. Mutation of REC114 L104 reduced the binding to both ANKRD31[27] and to TOPOVIBL (Supplementary Fig. 4b, lane 10). Similarly, the ANKRD31 W1842A mutation disrupted the interaction with REC114[27], as did the corresponding W562A mutation in TOPOVIBL (Fig. 2g, lane 9). As in our hands SUMO or MBP fusions of ANKRD31[1808−1857] aggregated unless bound to REC114, we could not determine its $K_d$ for REC114. To test for the mutually exclusive nature of the ANKRD31/TOPOVIBL binding to REC114, we performed ITC measurements of the interaction of REC114 with TOPOVIBL[452−579] in the absence or presence of saturating amounts of ANKRD31[1808−1857]. While TOPOVIBL[452−579] normally interacted with REC114 with a $K_d$ of 1.2 μM, we did not observe any binding when REC114 was pre-saturated with ANKRD31[1808−1857] (Supplementary Fig. 5f). Similarly, in pull-down assays, the interaction of REC114 with TOPOVIBL[452−479] was prevented when REC114 was first bound to ANKRD31[1808−1857] (Fig. 2k, lanes 8-11). These results strongly suggest that ANKRD31 prevents TOPOVIBL binding to the REC114 PH domain.

### In vivo analysis of the TOPOVIBL-REC114 interaction

To evaluate the biological significance of the interaction between the C-terminal region of TOPOVIBL and REC114, we generated mice that express mutant alleles of *Top6bl*, which are predicted to alter the interaction in vivo: i) *Top6bl*[W562A] harbouring the point mutation W562A that disrupt the interaction between TOPOVIBL and REC114 in vitro (Fig. 2); and ii) *Top6bl*[Δ17Ct] harbouring a truncation of the C-terminal helix of TOPOVIBL that interacts with REC114 by replacing the last 17 amino acids with 9 unrelated residues as a consequence of a frameshift immediately after W562 (Supplementary Fig. 6a–c). We verified by size exclusion chromatography of the C-terminal domain of TOPOVIBL (452-579) that the truncation of the last 17 amino acids did not significantly change the protein elution profile and thus did not induce major changes in its structure (Supplementary Fig. 6d). We also verified that both mutant proteins were expressed in mouse testes. As the TOPOVIBL signal is weak and only detected after immunoprecipitation, a quantitative assessment of protein levels in vivo is not possible (Supplementary Fig. 6e). In ovaries, *Top6bl* expression was detected by RT-PCR in both mutant mice (Supplementary Fig. 6f). Homozygous mutant mice were viable.

### *Top6bl* mutations lead to a DSB activity reduction in oocytes

We first investigated DSB formation during meiotic prophase of oocytes from embryonic ovaries (16 days post-coitum, dpc), when leptonema and zygonema are predominant[30]. To follow DSB

formation, we analysed the phosphorylated form of H2AX (γH2AX) that appears at chromatin domains around DSB sites upon DSB formation[31]. In wild-type oocytes, γH2AX was present over large chromatin domains at leptonema and zygonema and mostly disappeared at pachynema (Fig. 3a). In both *Top6bl* mutants, γH2AX intensity was strongly reduced at leptonema and zygonema: by 5.2- and 2.5-fold, respectively, in *Top6bl*[W562A/W562A], and by 8-fold and 7.6-fold, respectively, in *Top6bl*[Δ17Ct/Δ17Ct] oocytes, compared with wild-type oocytes (Fig. 3a, b). To determine whether this could be due to delayed DSB formation, we monitored γH2AX levels at a later developmental stage (18dpc) in *Top6bl*[Δ17Ct/Δ17Ct] mice and found a reduction by 7.8-fold (Supplementary Fig. 7a). These findings suggest that the two *Top6bl* mutations lead to a decrease of DSB activity.

We confirmed this hypothesis by quantifying DSB repair through the detection of DMC1 and RPA2. DMC1 binds to resected DSB ends and catalyses homologous strand exchange for DSB repair. RPA is recruited to resected DSB ends before and also after strand exchange during second-end capture for repair[2,32]. In wild-type oocytes, we detected DMC1 and RPA foci that colocalized with the chromosome axis at leptonema and zygonema. The number of DMC1 foci was decreased in both *Top6bl* mutants at leptonema (3.1-fold in *Top6bl*[W562A/W562A] and 18.5-fold in *Top6bl*[Δ17Ct/Δ17Ct] mice) (Fig. 3c, d). DMC1 level in *Top6bl*[Δ17Ct/Δ17Ct] mice was close to the background level because DMC1 foci at leptonema and zygonema were reduced by 22.9-fold in *Top6bl*[-/-] mice where DSB formation is abolished (Fig. 3d). The reduction of DMC1 foci was also observed at zygonema (Fig. 3d). We obtained similar results for RPA2 foci (Supplementary Fig. 7b, c). In wild-type oocytes, DSB repair promotes interactions between homologues that are stabilized by the recruitment and assembly of several proteins, including SYCP1, to form the synaptonemal complex (Fig. 3a)[33]. In both *Top6bl*[W562A/W562A] and *Top6bl*[Δ17Ct/Δ17Ct] mice, we observed only short stretches of synapsis and very few nuclei with full synapses. In 16dpc wild-type ovaries, 42.5%, 32.5% and 20% of oocytes were in leptonema, zygonema and pachynema (n = 315), respectively, compared with 61.9%, 37.7% and 0.4% (n = 496) in *Top6bl*[W562A/W562A], and 57.6%, 42.4% and 0% (n = 151) in *Top6bl*[Δ17Ct/Δ17Ct] ovaries. Overall, these results are consistent with a reduced DSB activity that affects synapsis formation between homologues.

As the formation of meiotic DSBs depends on the pre-DSB proteins IHO1, REC114, MEI4, ANKRD31 and MEI1[23,25−28,34], and because the two *Top6bl* mutations disrupt the interaction interface with REC114, we assessed REC114 cytological localization in *Top6bl*[W562A/W562A] and *Top6bl*[Δ17Ct/Δ17Ct] mice. In wild-type oocytes, REC114, IHO1, MEI4 and ANKRD31 form several hundred foci on chromosome axes at leptonema and they progressively disappear as DSBs form[24−28]. The number of REC114 axis-associated foci was significantly higher in *Top6bl*[W562A/W562A] and particularly in *Top6bl*[Δ17Ct/Δ17Ct] mice at leptonema and especially at zygonema compared with wild-type oocytes (Fig. 3e, f). This higher number of foci can be explained by a reduction of DSB activity. Indeed, it was previously shown that REC114 (and

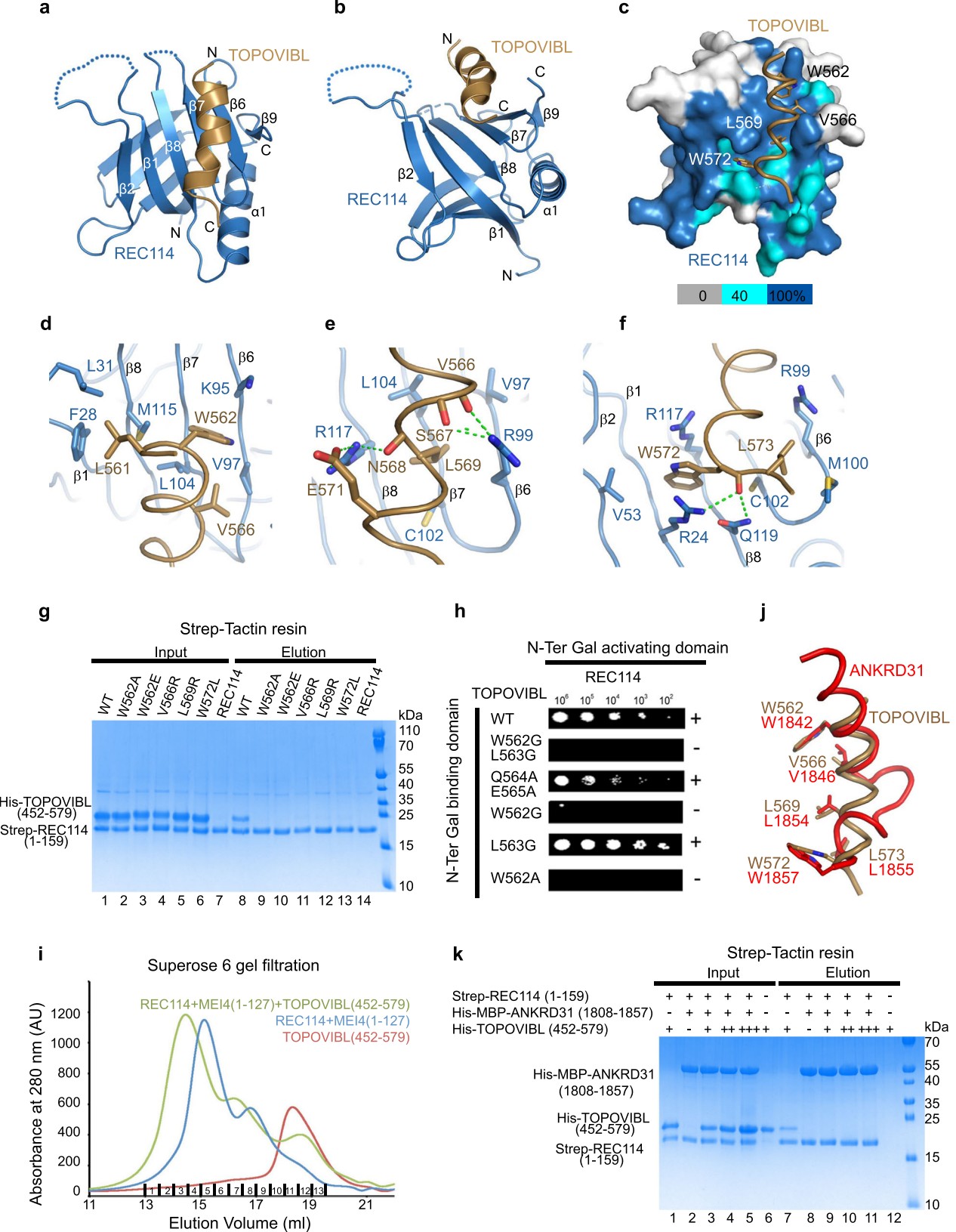

MEI4) foci are displaced from chromosome axes upon DSB formation in wild-type meiocytes, likely as a regulation to turn-off DSB activity, and accumulate in DSB-defective mutant spermatocytes such as Spo11[−/−][24,25]. This accumulation of REC114 at zygonema in the absence of meiotic DSB activity also applies to oocytes (Supplementary Fig. 7f)[25]. We obtained similar results for MEI4 and ANKRD31 foci

which were readily detectable on axes at both leptonema and zygonema but with an increased number mostly at zygonema in Top6bl[W562A/W562A] and Top6bl[Δ17Ct/Δ17Ct] oocytes (Supplementary Figs. 7d, e, 8a, b). Overall, these analyses demonstrate that the loading of REC114, MEI4 and ANKRD31 is not altered by the Top6bl[W562A] and Top6bl[Δ17Ct] mutations.

**Fig. 2 | Structure of the REC114-TOPOVIBL complex. a** Ribbon representation of the overall structure of the REC114-TOPOVIBL complex. The REC114 PH domain is in blue, and the TOPOVIBL peptide in brown. Alpha helices (α) and beta sheets (β) are labelled. **b** REC114-TOPOVIBL structure rotated 60° around the horizontal axis, compared to panel **a**. **c** Surface representation of REC114 to highlight the conserved surface residues. Sequence conservation is represented from grey to blue according to the colour scale bar below. The sequence alignment used for conservation calculation is shown in Supplementary Fig. 2c. TOPOVIBL is shown as a cartoon and the key interacting residues as sticks. **d** Details of the interaction between the N-terminal part of the TOPOVIBL helix (brown) and REC114 (blue). TOPOVIBL W562 inserts into a hydrophobic pocket on the β-sheet formed by strands β1 and β6-β8. **e** The central part the TOPOVIBL helix (brown) forms several hydrogen bonds (green dotted lines) with conserved REC114 residues. L569 interacts with another hydrophobic cavity formed by β6-β8. **f** The C-terminal W572 residue of TOPOVIBL forms hydrophobic and charged interactions with REC114. **g** Pull-down experiments of Strep-tagged REC114$^{1-159}$ with TOPOVIBL$^{452-579}$ mutants

indicated above the lanes. **h** Essential role of TOPOVIBL W562 in the interaction with REC114 shown by yeast two-hybrid assays. Full length TOPOVIBL (WT or with the indicated point mutations) was tested for interaction with full length REC114. Growth (+ or -) was assayed on medium lacking leucine, tryptophane and histidine and with 5 mM 3-amino-1,2,4-triazole. **i** Overlay of Superose 6 gel filtration elution profiles of Strep-REC114-MEI4$^{1-127}$, TOPOVIBL$^{452-579}$ and their complex. SDS-PAGE gels with eluted fractions are shown in Supplementary Fig. 5c. **j** The key TOPOVIBL and ANKRD31 residues that interact with the REC114 β-sheet (β1, β2, β6-β8) are in similar positions. ANKRD31 structure PDB code: 6NXF. **k** ANKRD31 prevents TOPOVIBL binding to REC114. His-TOPOVIBL$^{452-579}$, Strep-REC114$^{1-159}$ and Strep-REC114$^{1-159}$ bound to His-MBP-ANKRD31$^{1808-1857}$ were first purified by affinity chromatography and gel filtration. Proteins were mixed as indicated above the lanes. While TOPOVIBL$^{452-579}$ co-purifies with REC114$^{1-159}$ (lane 7), it does not co-purify even when used at two (++) or four (+++) times higher concentrations when REC114$^{1-159}$ is previously bound to His-MBP-ANKRD31 (lanes 10,11). Source data are provided as a Source Data file.

These meiotic prophase defects affected oogenesis: follicle number was strongly reduced in ovaries of $Top6bl^{W562A/W562A}$ and particularly $Top6bl^{Δ17Ct/Δ17Ct}$ mice (Supplementary Fig. 9a, b, d). Indeed, $Top6bl^{Δ17Ct/Δ17Ct}$ female mice were sterile, whereas $Top6bl^{W562A/W562A}$ female mice were sub-fertile (Supplementary Fig. 9c). We conclude that the TOPOVIBL-REC114 interaction is important for the normal level of meiotic DSB formation in oocytes, and that the truncation of the last 17 amino acids of TOPOVIBL leads to a stronger decrease of DSB activity as compared to the W562A mutation.

### In *Top6bl* spermatocytes, DSBs are delayed genome-wide and decreased in sub-telomeric regions

Unlike female meiosis, DSB activity was efficient in spermatocytes from both *Top6bl* mutants, as indicated by the detection and quantification of γH2AX at late leptonema and zygonema (Fig. 4a–c). Conversely, at early/mid leptonema, γH2AX levels were lower in $Top6bl^{W562A/W562A}$ and $Top6bl^{Δ17Ct/Δ17Ct}$ than wild-type spermatocytes, suggesting a delay in DSB formation (Fig. 4c).

DMC1 and RPA2 foci also appeared later in both mutants compared with wild-type spermatocytes (Fig. 4d–f and Supplementary Fig. 10b–d). The number of DMC1 and RPA2 foci was lower in $Top6bl^{W562A/W562A}$ and $Top6bl^{Δ17Ct/Δ17Ct}$ than in wild-type spermatocytes, particularly at early/mid leptonema (8.7- and 3.7-fold reduction of DMC1 foci and 11.7- and 7.4-fold reduction of RPA2 foci in $Top6bl^{W562A/W562A}$ and $Top6bl^{Δ17Ct/Δ17Ct}$, respectively, compared with wild-type spermatocytes). Conversely, at zygonema and pachynema, the number of DMC1 and RPA2 foci was similar in wild-type and mutant spermatocytes (Fig. 4e and Supplementary Fig. 10b). These findings suggest an efficient but delayed formation of DMC1 and RPA2 foci, and efficient DSB repair in $Top6bl^{W562A/W562A}$ and $Top6bl^{Δ17Ct/Δ17Ct}$ spermatocytes.

The localization of the DSB axis-associated proteins REC114, ANKRD31 and MEI4 at early/mid leptonema and the number of foci were similar or higher than in wild-type spermatocytes (Fig. 5a, b, d and Supplementary Fig. 11a–c). The number of foci gradually decreased from early/mid leptonema in wild-type spermatocytes, but only after late leptonema (ANKRD31) or after early/mid leptonema (REC114) and with a slower kinetic in the *Top6bl* mutants (Fig. 5c, e). These kinetic alterations are compatible with the observed delayed DSB activity because these axis-associated proteins disassemble from the axis upon DSB formation[25,27,28].

To directly evaluate DSB activity and to map DSB sites, we monitored DMC1 enrichment by DMC1 chromatin-immunoprecipitation (ChIP), followed by ssDNA enrichment (DMC1-Single Strand DNA Sequencing, SSDS)[35] in $Top6bl^{Δ17Ct/Δ17Ct}$ mice. We identified 16780 DSB hotspots in $Top6bl^{Δ17Ct/Δ17Ct}$, among which 13261 (79%) overlapped with wild-type hotspots (Fig. 6a). This indicated efficient PRDM9-dependent DSB localization in the mutant, as confirmed also by the

absence of significant signal at PRDM9-independent hotspots[36] (Supplementary Fig. 12). The hotspot intensity was similar in $Top6bl^{Δ17Ct/Δ17Ct}$ and wild-type samples for most hotspots (Fig. 6b). 3519 peaks were however specific to $Top6bl^{Δ17Ct/Δ17Ct}$ mice (Fig. 6a). The average intensity of the DMC1-SSDS signal of the 3519 $Top6bl^{Δ17Ct/Δ17Ct}$-specific peaks was weak (Supplementary Fig. 13a upper panel). But interestingly these regions also show some DSB activity in wild-type, although with a -1.7-fold lower average DMC1-SSDS signal as compared to $Top6bl^{Δ17Ct/Δ17Ct}$ mice (Supplementary Fig. 13a upper panel). At these sites, the H3K4me3 enrichment detected in wild-type mice of the same *Prdm9* genotype ($Prdm9^{Dom2}$ from B6 strain) but not from mice expressing the $Prdm9^{Cst}$ variant (RJ2 strain) confirms that they are *Prdm9*-dependent hotspots (Supplementary Fig. 13a lower panel). This indicates that these 3519 peaks are not new hotspots specific to the $Top6bl^{Δ17Ct/Δ17Ct}$ mutant context but that their detection in this mutant is due to the increased DMC1-SSDS signal. We asked if we could also detect any increase of DMC1-SSDS signal in $Top6bl^{Δ17Ct/Δ17Ct}$ among the subset of the weakest hotspots identified in wild-type, but this was not the case (bin 1 from Supplementary Fig. 13b). It is therefore possible that these 3519 sites have some specific feature leading to their differential activity in $Top6bl^{Δ17Ct/Δ17Ct}$ *vs* wild-type. A differential analysis of DMC1-SSDS signal intensity at common hotspots (using the DESeq2 R package, see Methods) showed that in $Top6bl^{Δ17Ct/Δ17Ct}$ mice, most of the hotspots (86%) were of similar intensity compared to wild-type, and that the signal was decreased (by 1.2- to 24-fold) in 12% of hotspots, and increased (by 1.45- to 3.3-fold) in 2% of hotspots (green and red dots respectively, Fig. 6c). Visual analysis highlighted that in $Top6bl^{Δ17Ct/Δ17Ct}$ mice, the DMC1-SSDS signal was decreased at hotspots located near the q-arm telomeres, the telomeres distal to centromeres (Fig. 6d and Supplementary Figs. 14, 15). We quantified this sub-telomeric phenotype by different approaches. First, by quantifying the ratio of the mutant/wild-type signal along the chromosome arms, we observed a ≥ 2-fold decrease in the last few megabases proximal to the q-arm telomeres (Fig. 6e). Second, in the ten hotspots closest to the q-arm telomeres, located within about 4 Mb of the telomere, DMC1-SSDS signal intensity at most hotspots for each chromosome was lower in $Top6bl^{Δ17Ct/Δ17Ct}$ than wild-type samples (blue for autosomes, red for the X chromosome) (Fig. 6b). Third, quantification of this effect at each chromosome showed a reduced DMC1-SSDS signal within the ten q-arm telomeric adjacent hotspots in $Top6bl^{Δ17Ct/Δ17Ct}$ mice at most chromosomes (Fig. 6f and Supplementary Fig. 13c). Fourth, using DESeq, we evaluated the effect of the distance from telomeres, and found that the effect was most pronounced at hotspots located within 3 Mb from telomeres and decreased when we tested larger intervals (Fig. 6g and Supplementary Fig. 13d). In this analysis, the statistical significance also depended on the hotspot number in the tested region for each chromosome (see, for instance, chromosome 17, Supplementary Fig. 13e). We conclude that in $Top6bl^{Δ17Ct/Δ17Ct}$ mice, the

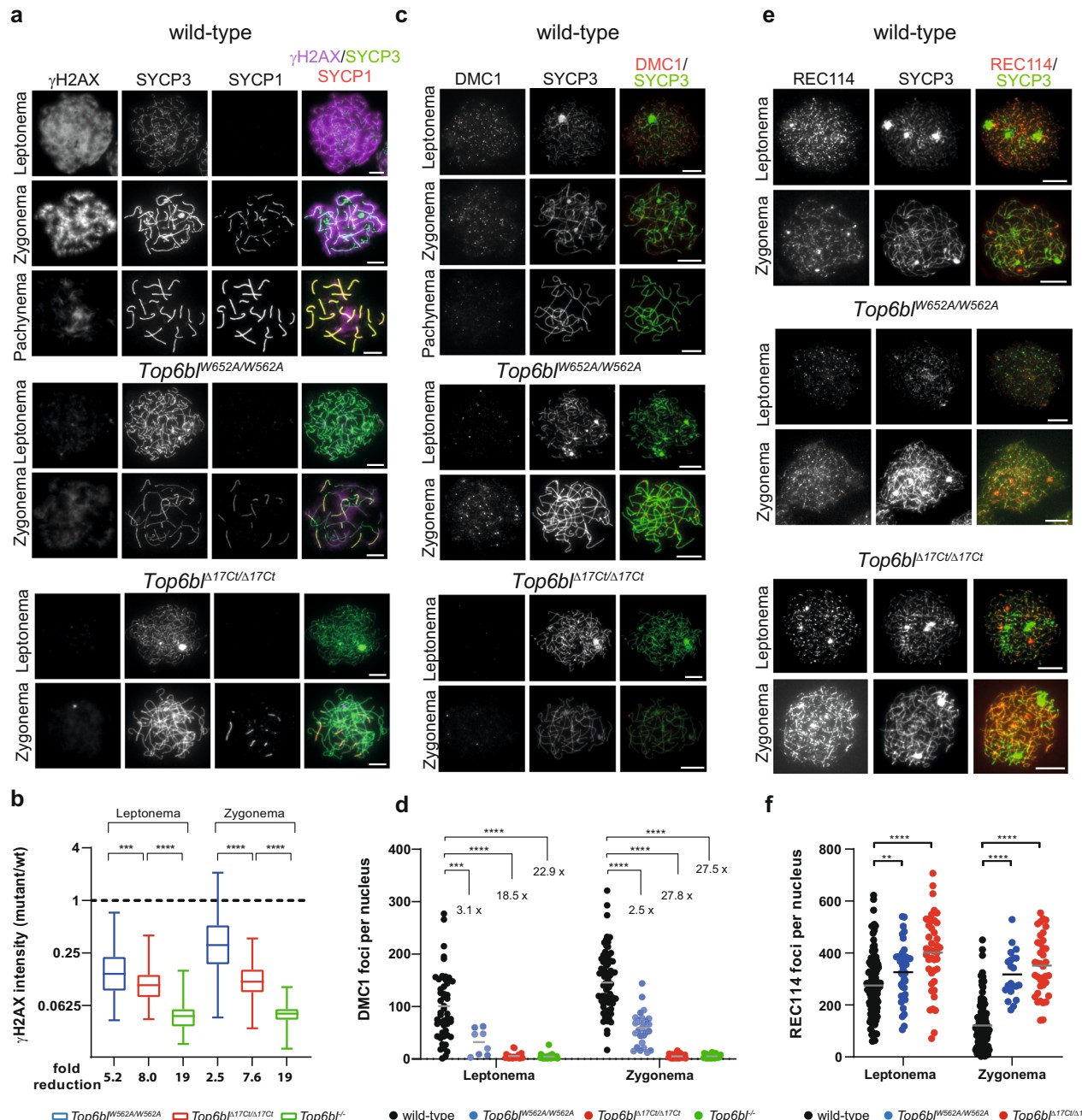

**Fig. 3 | In *Top6bl^(W562A/W562A)* and *Top6bl ^(Δ17Ct/Δ17Ct)* mice, meiotic DSB activity is decreased in oocytes. a** Immunostaining of γH2AX, SYCP3 and SYCP1 in oocytes from 16 dpc wild-type (+/+), *Top6bl^(W562A/W562A)*, and *Top6bl^(Δ17Ct/Δ17Ct)* ovaries. Scale bar, 10 μm. **b** Quantification of γH2AX signal intensity in leptotene and zygotene (or zygotene-like) nuclei of oocytes from 16 dpc wild-type (+/+ or *Top6bl^(+/Δ17Ct)*), *Top6bl^(W562A/W562A)*, *Top6bl^(Δ17Ct/Δ17Ct)*, and *Tob6bl^(-/-)* mice (*n* = 5, 2, 2, and 1 mouse/geno-type). Number of nuclei at leptonema: 126, 123, 73, and 23; number of nuclei at zygonema: 241, 132, 64, and 33 for each genotype. Ratios of the integrated intensity between the mean values in mutant nuclei and in wild-type nuclei are plotted (box plot as defined in Methods). *P* values were determined using the two-tailed unpaired Mann-Whitney test. The fold reduction is the ratio of the wild-type to mutant mean values. **c** Immunostaining of DMC1 and SYCP3 in oocytes from 16 dpc or 17 dpc wild-type (+/+), *Top6bl^(W562A/W562A)* and *Top6bl^(Δ17Ct/Δ17Ct)* ovaries. Scale bar, 10 μm. **d** Quantification of DMC1 foci. DMC1 axis-associated foci were counted in

leptotene and zygotene nuclei of oocytes from 16 and 17 dpc wild-type (+/+ or *Top6bl^(+/W562A)*), *Top6bl^(W562A/W562A)*, *Top6bl^(Δ17Ct/Δ17Ct)*, and *Top6bl^(-/-)* mice. Number of nuclei at leptonema: 50, 8, 22, and 28; number of nuclei at zygonema: 79, 28, 29, and 27 for each genotype. Grey bars show the mean values. P values were determined using the two-tailed unpaired Mann-Whitney test. The fold difference relative to wild-type is shown. **e** Immunostaining of REC114 and SYCP3 in oocytes from 16dpc wild-type (+/+), *Top6bl^(W562A/W562A)*, and *Top6bl^(Δ17Ct/Δ17Ct)* ovaries. Scale bar, 10 μm. **f** Quantification of axis-associated REC114 foci in leptotene and zygotene oocytes from 15 dpc wild-type (+/+), *Top6bl^(W562A/W562A)*, and *Top6bl^(Δ17Ct/Δ17Ct)* mice (*n* = 1 mouse/genotype). Number of nuclei at leptonema: 51 and 43; number of nuclei at zygonema: 58 and 39 in wild-type and *Top6bl^(Δ17Ct/Δ17Ct)* oocytes, respectively. Grey bars show the mean values. *P* values were determined using the two-tailed unpaired Mann-Whitney test. Source data are provided as a Source Data file.

DMC1-SSDS signal is specifically reduced in the 3 Mb sub-telomeric region of the q-arm of most chromosomes. This finding could indicate a DSB decrease or a higher DMC1 turnover. This region-specific alteration in *Top6bl^(Δ17Ct/Δ17Ct)* mice may not affect homologous

interactions between autosomes because these interactions should be ensured by DSB sites along the chromosome arms which level of activity is overall similar in *Top6bl^(Δ17Ct/Δ17Ct)* and wild-type mice. However, on the X and Y chromosomes, which depend on recombination in

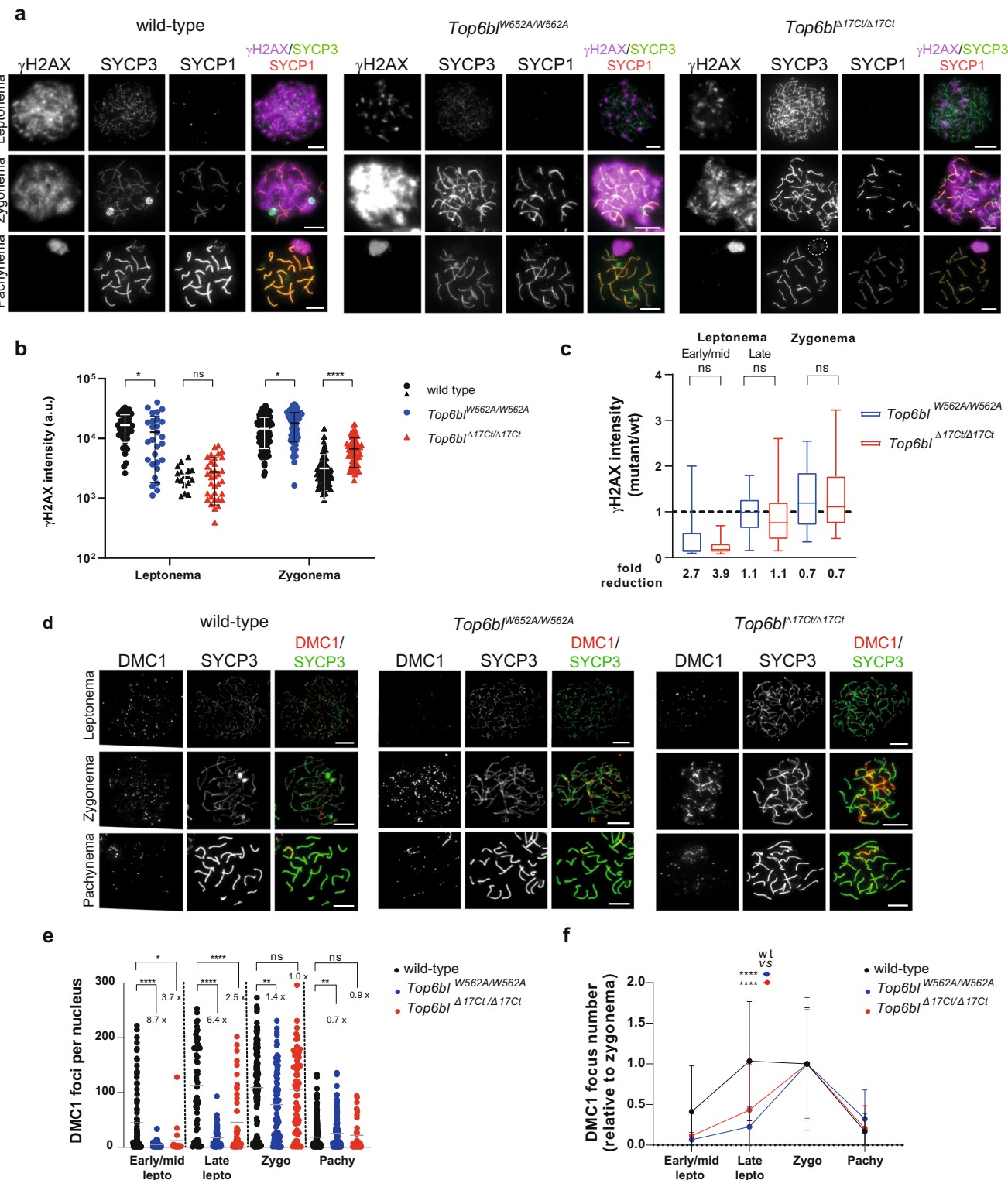

the PAR, and which is located proximal to the telomere, the decreased DSB activity in this region in *Top6bl*[Δ17Ct/Δ17Ct] mice (Fig. 6d) could influence their homologous interaction.

We tested this possibility by monitoring synapsis and bivalent formation on autosomes and on the X and Y chromosomes. Synapsis formation was normal on autosomes at pachynema in both mutants (Supplementary Table 4). However, the X and Y chromosomes were frequently unsynapsed in pachytene nuclei (41% of *Top6bl*[W562A/W562A], 72% of *Top6bl*[Δ17Ct/Δ17Ct] *vs* 18% in wild-type) (Fig. 7a). At metaphase, an increased frequency of nuclei with 21 DAPI staining bodies (Fig. 7b, d)

and a decreased number of XY bivalent (Fig. 7c, e) was detected in both *Top6bl* mutants. We conclude that the decreased DMC1 signal on the X chromosome sub-telomeric region is compatible with decreased DSB activity rather than with rapid DMC1 turnover. We propose that the TOPOVIBL-REC114 interaction is specifically required at sub-telomeric regions for full DSB activity and therefore, is essential for X/Y chromosome synapsis and segregation.

These molecular and cytological phenotypes should have specific consequences on meiotic prophase and downstream events, because unsynapsed X/Y chromosomes induce metaphase I arrest[37,38]. Indeed,

**Fig. 4 | Efficient but delayed DSB formation in *Top6bl* mutant spermatocytes.**
**a** Immunostaining of γH2AX, SYCP3 and SYCP1 in spermatocytes from 12 dpp wild-type (+/+), *Top6bl*[W562A/W562A], and *Top6bl*[Δ17Ct/Δ17Ct] mice. A white dotted circle highlights the unsynapsed X and Y chromosomes at pachynema (SYCP3 staining) in *Top6bl*[Δ17Ct/Δ17Ct] mice. Scale bar, 10 μm. **b** Quantification of γH2AX intensity (mean ± SD; a.u., arbitrary units) in leptotene and zygotene spermatocytes from 12 dpp wild-type, *Top6bl*[W562A/W562A], and *Top6bl*[Δ17Ct/Δ17Ct] mice (*n* = 1 mouse per genotype). Each mutant had a wild-type control tested in parallel (labelled with black spheres or triangles). Number of nuclei: 44, 28, 17, and 34 at leptonema; 132, 61, 77, and 66 at zygonema. *P* values were determined using the two-tailed unpaired Mann-Whitney test. **c** Quantification of γH2AX intensity in *Top6bl*[W562A/W562A] and *Top6bl*[Δ17Ct/Δ17Ct] spermatocytes relative to wild-type at early/mid, late leptotene, and zygotene (box plot as defined in Methods). Stages were defined as described in Methods (see Supplementary Fig. 10a). Number of nuclei: 19, 201 at early/mid, 33, 290 at late leptotene, and 88, 333 at zygotene in *Top6bl*[W562A/W562A] and *Top6bl*[Δ17Ct/Δ17Ct] mice, respectively. The fold reduction is the ratio of the mean values in wild-type and mutant samples. *P* values were determined using the two-tailed unpaired Mann-

Whitney test. **d** Immunostaining of DMC1 and SYCP3 in spermatocytes from 14 dpp wild-type (+/+), *Top6bl*[W562A/W562A], and *Top6bl*[Δ17Ct/Δ17Ct] mice. Scale bar, 10 μm.
**e** Quantification of DMC1 foci. Axis-associated DMC1 foci were counted in leptotene (early/mid and late), zygotene, and pachytene nuclei of spermatocytes from wild-type (+/+), *Top6bl*[W562A/W562A], and *Top6bl*[Δ17Ct/Δ17Ct] mice (*n* = 3 wild-type, and *n* = 2 mice per mutant genotype). Number of nuclei: 95, 50 and 18 at early/mid leptotene, 55, 63 and 45 at late leptotene, 120, 92 and 66 at zygotene, and 261, 207 and 36 at pachytene for each genotype, respectively. Grey bars show the mean values. *P* values were determined using the two-tailed unpaired Mann-Whitney test.
**f** Variation of DMC1 focus number during prophase. For each genotype (wild-type, *Top6bl*[W562A/W562A], and *Top6bl*[Δ17Ct/Δ17Ct]), the number of DMC1 foci at the indicated stages (dataset as in panel e) was normalized to the mean number at zygonema (set to 1). Mean values ± SD are shown. Statistical significance between wild-type and each mutant (blue *Top6bl*[W562A/W562A]; red *Top6bl*[Δ17Ct/Δ17Ct]) was tested at late leptonema using the two-tailed unpaired Mann-Whitney test. Source data are provided as a Source Data file.

*Top6bl*[W562A/W562A] and *Top6bl*[Δ17Ct/Δ17Ct] spermatocytes proceed through prophase like wild-type spermatocytes, but they arrested in metaphase and many cells were apoptotic, particularly in the *Top6bl*[Δ17Ct] mutant. This metaphase I arrest was correlated with reduced sperm production in both mutants and fertility loss in *Top6bl*[Δ17Ct/Δ17Ct] mice (Supplementary Fig. 9e–k).

## Discussion

A central question after the identification of the axis-associated proteins essential for meiotic DSB formation is to understand their function. Here, we found that TOPOVIBL CTD directly interacts with the REC114 PH domain, and identified residues required for this interaction in vitro.

We propose that TOPOVIBL can bind simultaneously to SPO11 through its transducer domain and to REC114 through the CTD. Both mouse TOPOVIBL and SPO11 structures have been modelled by the AlphaFold2 structure prediction program[29] with high predicted accuracy (AlphaFold2 Protein Structure Database code- AF-J3QMY9 and AF-Q9WTK8). We further used AlphaFold2[29] to model the SPO11/TOPO-VIBL complex and combined this model with the crystal structure of the REC114-PH/TOPOVIBL[559-576] complex determined in this study (Fig. 8a and Supplementary Fig. 16d, e). The model gave very high or confident pLDDT scores for most residues in the GHKL-like and transducer domains of TOPOVIBL as well as in 5Y-CAP and Toprim domains of SPO11 (Supplementary Fig. 16a, b, e, g–l, respectively). The predicted aligned errors are low except for the relative positions of the transducer domains of each monomer (Supplementary Fig. 16c, f). Compared to the TopoVIB structure from archaea[39], the putative mouse TOPOVIBL ATP binding site is predicted to be degenerated with the lack of the ATP-lid structure and its replacement by a helical insertion of the transducer domain (Supplementary Fig. 8b). Indeed, TOPOVIBL contains mutations within the conserved Bergerat fold motifs[5,7] essential for ATP binding and hydrolysis[40]. TOPOVIBL is thus predicted not to bind ATP. TOPOVIBL also lacks both the N-terminal "strap" region involved in dimerization and the H2TH domain. Thus, the ATP mediated dimerization, observed for archaeal TopoVIB, is unlikely to occur in the case of TOPOVIBL. SPO11 however can be modelled as homodimer and its catalytic site, formed by the two protomers is predicted to be equivalent to the one described for TopoVIA[41] (Fig. 8d, e and Supplementary Fig. 16h, i). Finally, the modelled SPO11-TOPOVIBL interface involving the three conserved helices from TOPOVIBL transducer domain and part of the 5Y-CAP domain of SPO11 starting with the second helix (44-63), resembles the archaeal TopoVI complex but with an additional helix of the transducer domain (Fig. 8c and Supplementary Fig. 16g). The REC114-TOPOVIBL complex, as characterized in this study, is connected by the flexible region of the TOPOVIBL CTD to the SPO11/TOPOVIBL core (Fig. 8a).

Although the exact REC114 effect on SPO11/TOPOVIBL organization/conformation is unknown, REC114 binding to TOPOVIBL could stabilize the SPO11/TOPOVIBL complex dimerization. This hypothesis is supported by the stoichiometry (2:1) of the Rec114/Mei4 complex in *S. cerevisiae*[15] and by a predicted heterotrimer from *C. elegans* DSB-1/DSB-2/DSB-3 complex[42], where DSB-1 and −2 are predicted REC114 orthologs[43,44] and DSB-3, the MEI4 ortholog[45]. Our observation that REC114 could bind simultaneously to TOPOVIBL and MEI4 (Fig. 2j) is compatible with a model where MEI4/REC114 would act as clamp to hold the TOPOVIBL/SPO11 complex. Nevertheless, in mice, REC114 interactions are certainly more complex because its PH domain also interacts with ANKRD31[27], thus likely competing with TOPOVIBL (Fig. 2k and Supplementary Fig. 5f). Moreover, REC114 interacts with IHO1 in yeast-two hybrid assays[26]. The interplay between these different interactions remains to be determined and disrupting the interaction between TOPOVIBL and REC114 may have additional consequences on other interactions.

The interaction between the C-terminal motif of TOPOVIBL and the PH domain of REC114 is predicted to be conserved in many metazoans based on the presence of the motif and the PH domain (Fig. 1f[7,23,25,27]). TOPOVIBL is found outside from metazoans but is highly divergent and no eukaryotic consensus motif can be identified at its C-terminal end[7]. With the contribution of genetics, both TOPO-VIBL and REC114 homologs have also been identified in fungi and green plants[5,23]. In *S. cerevisiae*, TOPOVIBL appears to be split into two proteins Rec102 and Rec104. Phylogenetic and biochemical analysis identified Rec102 as homologous to the transducer domain of TOPO-VIBL, while Rec104 is predicted to be at the position of the GHKL domain[5,15]. *S. cerevisiae* Rec114 has the conserved PH domain interacting with both, Rec102 and Rec104 in yeast-two hybrid assays and the Rec114 residues involved in the interaction are essential for DSB formation[15,16]. According to the AlphaFold2 model of *S. cerevisiae* Rec114 (AF-A0A6V8S448), many of the residues of the mouse REC114 PH domain required for the interaction with mouse TOPOVIBL are not conserved in *S. cerevisiae*. These include mouse REC114 V97 and L104, shown in this study as essential for binding to TOPOVIBL (Supplementary Fig. 4b). In addition, the hydrophobic pocket of REC114 accommodating TOPOVIBL W562 (Fig. 2d) seems absent in *S. cerevisiae* Rec114. Thus, we conclude that a potentially equivalent regulation of DSB activity could be mediated by *S. cerevisiae* Rec114 but with distinct molecular interactions. The regulatory function of *S. cerevisiae* Rec114 for DSB activity is also illustrated by the Mec1/Tel1 (ATR/ATM) dependent phosphorylation of Rec114 and its consequence on down-regulating DSB activity[46], a modification and regulation also observed for *C. elegans* DSB-1[42]. In *A. thaliana*, both the TOPOVIBL ortholog, MTOPVIB[6], and REC114[23] have been identified. Although the PH domain of REC114 is conserved[27], the metazoan C-terminal motif is

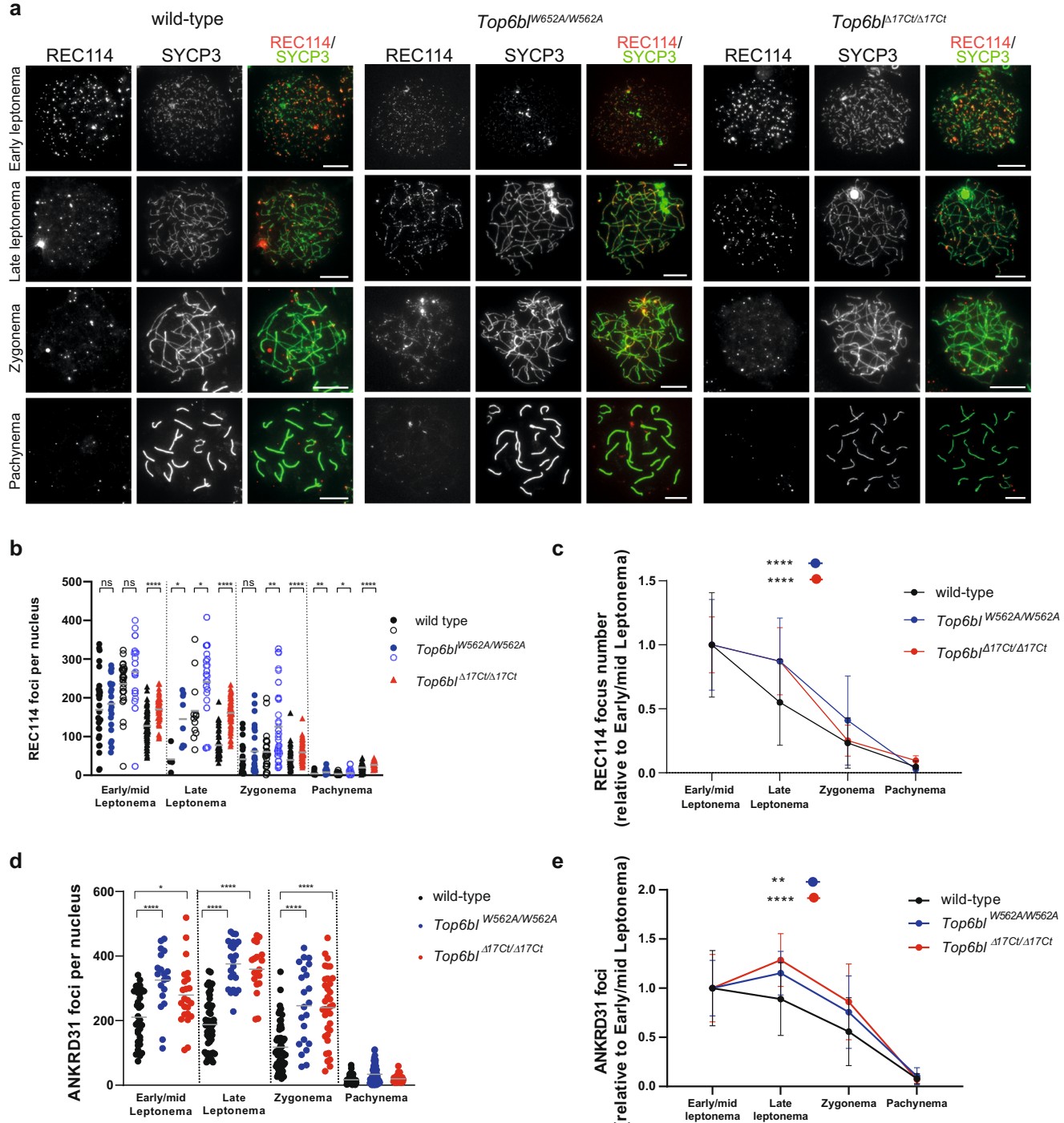

**Fig. 5 | Efficient REC114 and ANKRD31 loading in *Top6bl* mutant spermatocytes.** **a** Immunostaining of REC114 and SYCP3 in spermatocytes from 12-14dpp wild-type (+/+), *Top6bl*^*W562A/W562A*^, and *Top6bl*^*Δ17Ct/Δ17Ct*^ mice. Scale bar, 10 μm. **b** Quantification of axis-associated REC114 foci in early/mid and late leptotene, zygotene and pachytene spermatocytes from 12-14dpp wild-type (+/+ or *Top6bl*^*+/Δ17Ct*^: *n* = 4), *Top6bl*^*W562A/W562A*^ (*n* = 2), and *Top6bl*^*Δ17Ct/Δ17Ct*^ (*n* = 2) mice. Each mutant is compared with a wild-type control in parallel to control for variations between experiments. Results from *Top6bl*^*W562A/W562A*^ are from two independent experiments (labelled with filled and open circles respectively). Number of nuclei: wild-type: 31 early/mid L, 4 late L, 26 Z, 51 P; *Top6bl*^*W562A/W562A*^: 22 early/mid L, 8 late L, 26 Z, 24 P; wild-type 28 early/mid L, 12 late L, 24 Z, 46 P; *Top6bl*^*W562A/W562A*^ 18 early/mid L, 21 late L, 32 Z, 38 P; wild-type: 84 early/mid L, 59 late L, 84 Z, 75 P; *Top6bl*^*Δ17Ct/Δ17Ct*^: 76 early/mid L, 68 late L, 79 Z, 61 P. Grey bars show the mean values. P values were determined using the two-tailed unpaired Mann-Whitney test. **c** Variation of REC114 foci during prophase. The number of foci at early/mid leptonema, late leptonema, zygonema and pachynema (dataset as in panel b) relative to the mean number at early/mid

leptonema was plotted for wild-type, *Top6bl*^*W562A/W562A*^, and *Top6bl*^*Δ17Ct/Δ17Ct*^ mice. The mean values ± SD are shown. Statistical significance was tested at late leptonema using the two-tailed unpaired Mann-Whitney test. **d** Quantification of axis-associated ANKRD31 foci in early/mid, late leptotene, zygotene and pachytene spermatocytes from 12-14 dpp wild-type (+/+ or *Top6bl*^*+/Δ17Ct*^: *n* = 2), *Top6bl*^*W562A/W562A*^ (*n* = 1), and *Top6bl*^*Δ17Ct/Δ17Ct*^ (*n* = 1) mice. Mean number of nuclei per genotype at early/mid leptotene (37, 19, and 23), late leptotene (51, 23, and 19), zygotene (61, 22, and 35), and pachytene (100, 61, and 31). Grey bars show the mean values. P values were determined using the two-tailed unpaired Mann-Whitney test. **e** Variation of ANKRD31 foci during prophase. The number of foci at early/mid leptonema, late leptonema, zygonema, and pachynema (dataset as in panel **d**) relative to the mean number at early/mid leptonema (set at 1) was plotted for wild-type, *Top6bl*^*W562A/W562A*^, and *Top6bl*^*Δ17Ct/Δ17Ct*^ mice. The mean values ± SD are shown. Statistical significance was tested at late leptonema using the two-tailed unpaired Mann-Whitney test. Source data are provided as a Source Data file.

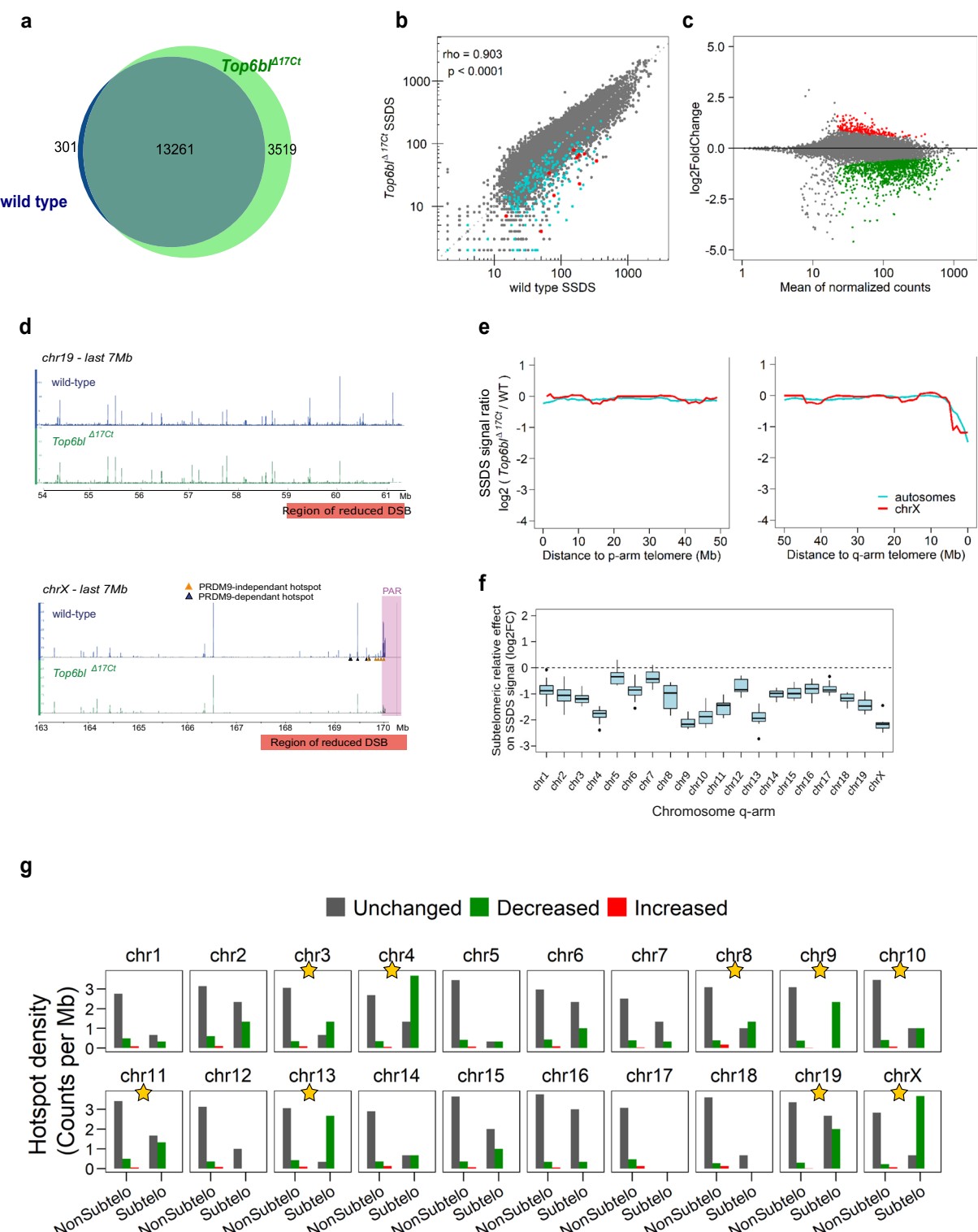

absent in MTOPVIB. It is therefore possible that the function of these proteins is mediated by distinct molecular interactions as suggested by interaction and protein localization assays in *A. thaliana*[47]. These divergences are correlated with a distinct role for *A. thaliana* REC114 which is dispensable for DSB formation[47]. AtREC114 may however regulate DSB activity at a distinct level, and it would be interesting to know if it plays a role in controlling the timing of DSB activity, since this is one of the REC114 roles we have uncovered in male mice.

The observation of meiotic DSB defects in female and male mice harbouring *Top6bl* mutations indicates that in vivo REC114 acts by directly interacting with the SPO11/TOPOVIBL complex. This interaction requires W562, a highly conserved residue of TOPOVIBL in metazoans (Fig. 1f)[7]. The stronger phenotype of mice carrying the *Top6bl^{Δ17Ct}* mutation (compared with *Top6bl^{W562A}*) could be explained by a lack of interaction between TOPOVIBL Δ17Ct and REC114 and a remaining weak interaction between TOPOVIBL W562A and REC114.

**Fig. 6 | Proper DMC1-SSDS signal localisation and intensity, excepted for q-arm sub-telomeric regions in *Top6bl*^(Δ17Ct/Δ17Ct) spermatocytes. a** In *Top6bl*^(Δ17Ct/Δ17Ct)spermatocytes, almost all wild-type hotspots and 20% of new hotspots are detected. **b** DMC1-SSDS signal correlation between wild-type and *Top6bl*^(Δ17Ct/Δ17Ct) mice at wild-type hotspots. The Spearman rho and associated p-value (two-sided) are shown. Ten telomere-proximal hotspots are highlighted for each autosome (blue) and for the X chromosome (red). In the PAR, only part of the DMC1-SSDS signal, which covers a large domain (see panel d), is included within hotspots. **c** MA plot of the DMC1-SSDS signal in *Top6bl*^(Δ17Ct/Δ17Ct) compared with wild-type samples. The log2 of the ratios was calculated using DESeq2. Hotspot with significantly increased or decreased signal are highlighted in red (*n* = 214) and green (*n* = 1098), respectively (adjusted *p* value < 0.1). Unchanged hotspots are in grey (*n* = 8043). The mean normalized count corresponds to the baseMean value from the DESeq2 analysis. **d** DSB maps of the 7 Mb telomere-proximal regions of the chromosomes 19 and X in wild-type (blue) and *Top6bl*^(Δ17Ct/Δ17Ct) (green) mice. On chromosome X PRDM9-independent and -dependent hotspots as defined by analysis in *Prdm9*^(-/-) mice[36] are identified by orange and blue triangles, respectively. The PAR is highlighted with a pink rectangle. All chromosome ends are shown in Supplementary Figs. 14, 15.

**e** DMC1-SSDS signal intensity decreases in *Top6bl*^(Δ17Ct/Δ17Ct) samples relative to wild-type samples in the q-arm sub-telomeric region (right panel). The DMC1-SSDS signal ratio within hotspots (log2-fold change estimated by DESeq2) was computed over 5Mb-windows with a 1Mb-step. The same analysis was performed in the 50 Mb adjacent to p-arm telomeres (left panel). **f** The q-arm sub-telomeric region effect in *Top6bl*^(Δ17Ct/Δ17Ct) mice. The averaged DMC1-SSDS signal ratio (*Top6bl*^(Δ17Ct/Δ17Ct)/wild-type) of the last ten hotspots of a given chromosome was compared to the averaged DMC1-SSDS signal ratio of ten randomly chosen, non-telomeric consecutive hotspots in the same chromosome. Box plots (as defined in Methods) represents the log2 fold-change (FC) between these values for ten randomizations. The control is shown in Supplementary Fig. 13c. **g** Decreased (green), increased (red), or unchanged (grey) hotspot density within the 3 Mb sub-telomeric (Subtelo) region relative to the nonsub-telomeric regions (NonSubtelo). Decreased, increased and unchanged hotspots were determined from the DESeq2 analysis, as shown in panel c. The densities of decreased and unchanged hotspots in the two studied regions were compared using the Pearson's Chi-square test, and p-values were adjusted for multiple testing using the Benjamini & Yekutieli method. Yellow stars indicate *p* value < 0.05. The Chi-square test results are provided in Supplementary Table 2.

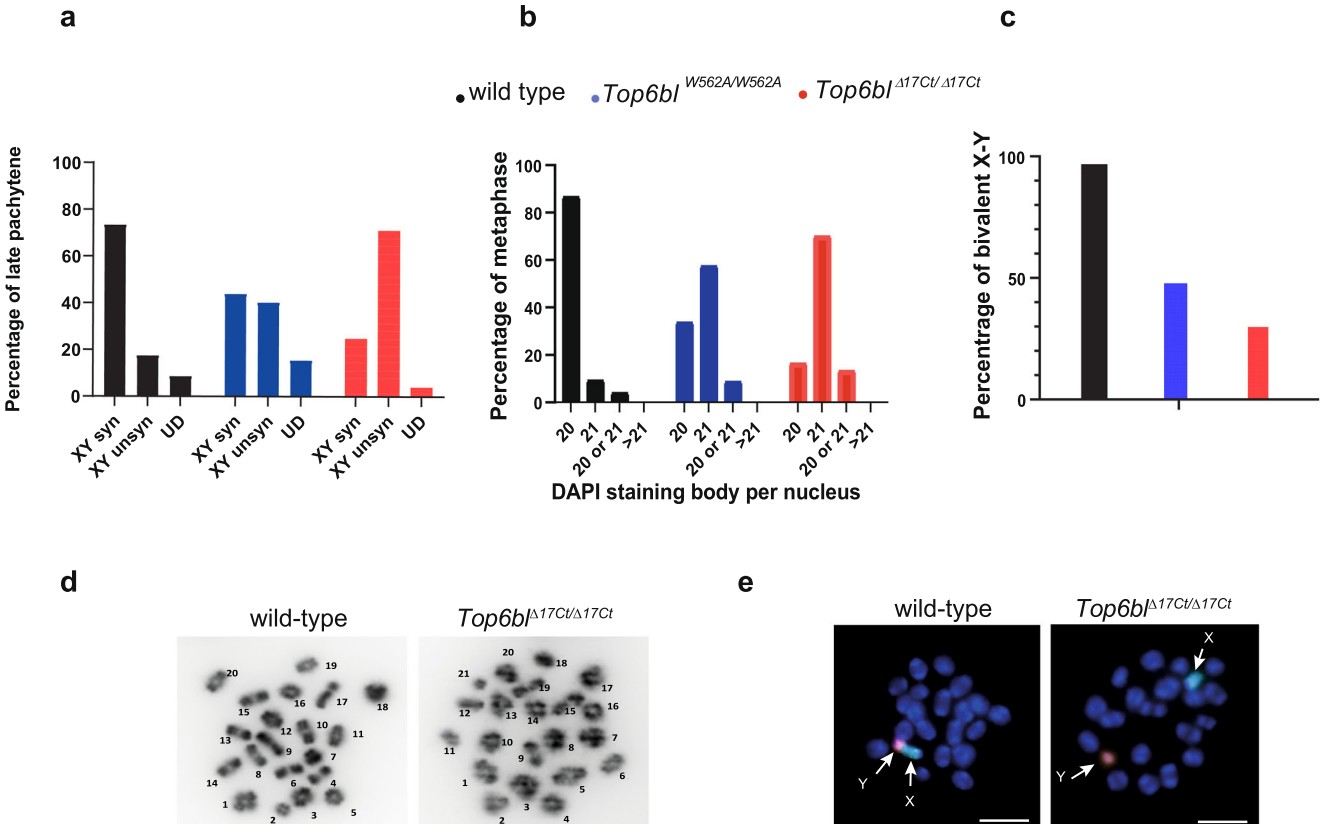

**Fig. 7 | Defective XY chromosome synapsis in *Top6bl* mutants. a** Synapsis quantification between the X and Y chromosomes at pachynema. Synapsis formation was monitored on spreads from late pachytene spermatocytes from adult wild-type (+/+, *Top6bl*^(+/W562A) or *Top6bl*^(+/Δ17Ct)), *Top6bl*^(W562A/W562A), and *Top6bl*^(Δ17Ct/Δ17Ct) mice, identified by staining with γH2AX, SYCP3 and SYCP1. Number of nuclei: wild-type (227), *Top6bl*^(W562A/W562A) (108), *Top6bl*^(Δ17Ct/Δ17Ct) (171). syn: synapsed; unsyn: unsynapsed; UD: undefined. The synapsed/unsynapsed ratios were significantly different between wild-type and *Top6bl*^(W562A/W562A) and *Top6bl*^(Δ17Ct/Δ17Ct) spermatocytes (Pearson's Chi-Square, 26.31 and 115.15, respectively). **b** Quantification of bivalent formation at metaphase I. Percentage of metaphases with 20, 21, 20 or 21, or >21 DAPI-stained bodies per nucleus from adult wild-type (*Top6bl*^(+/Δ17Ct)), *Top6bl*^(W562A/W562A), and *Top6bl*^(Δ17Ct/Δ17Ct) mice (*n* = 2 mice per genotype). Number of nuclei: 75, 80, and 97, respectively. The number of nuclei with 20 and 21 bivalents was significantly different between wild-type and *Top6bl*^(W562A/W562A) and between wild-type and *Top6bl*^(Δ17Ct/Δ17Ct) metaphases (Pearson's Chi-Square: 44.4 and 78.8

respectively). **c** Quantification of X and Y bivalents at metaphase I. The X and Y chromosomes were detected by FISH in metaphase spreads from adult wild-type (*Top6bl*^(+/Δ17Ct)), *Top6bl*^(W562A/W562A), and *Top6bl*^(Δ17Ct/Δ17Ct) mice (*n* = 2 mice per genotype). Number of nuclei: 84, 144, and 164 for wild-type, *Top6bl*^(W562A/W562A) and *Top6bl*^(Δ17Ct/Δ17Ct), respectively. The percentages of metaphase spreads with XY bivalents were significantly different between wild-type and *Top6bl*^(W562A/W562A), and between wild-type and *Top6bl*^(Δ17Ct/Δ17Ct) (Chi-Square Pearson: 54.2 and 96.9, respectively). **d** Representative images of DAPI-stained wild-type and *Top6bl*^(Δ17Ct/Δ17Ct) metaphase spreads. Scale bar, 10 μm. DAPI-stained bodies are numbered (arbitrarily): 20 are observed in wild-type, and 21 in *Top6bl*^(Δ17Ct/Δ17Ct) metaphase spreads (samples from panel b). **e** Representative images of a wild-type spermatocyte nucleus with the X and Y chromosomes forming a bivalent (left) and of a *Top6bl*^(Δ17Ct/Δ17Ct) spermatocyte nucleus with separated X and Y chromosomes (right)(samples from panel c). Blue, nuclei (DAPI staining); green, X chromosome probe; red, Y chromosome probe. Scale bar, 10 μm.

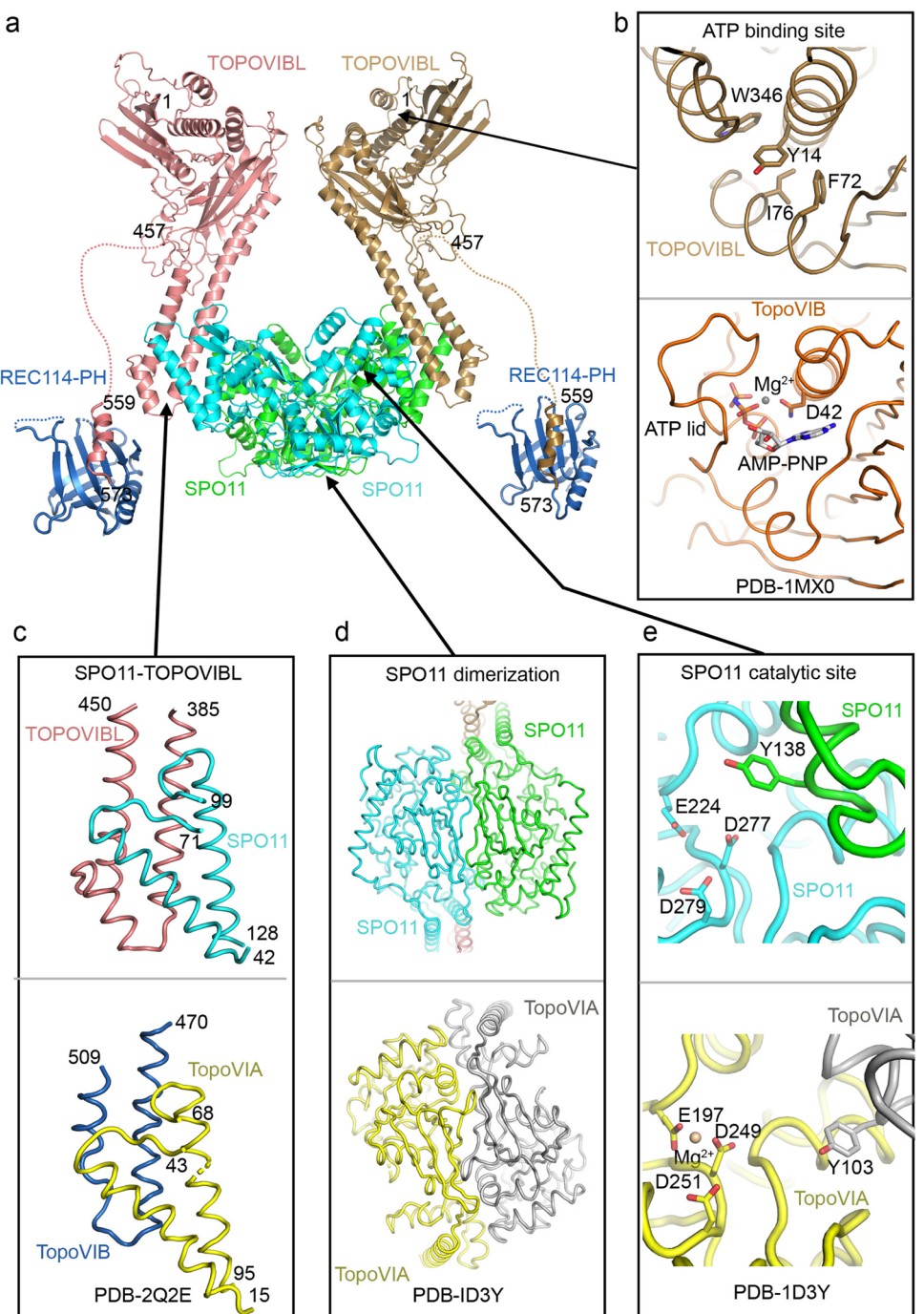

**Fig. 8 | Model of the SPO11-TOPOVIBL-REC114 complex. a** The structure of the complex of SPO11 and the C-terminal part of the TOPOVIBL transducer domain (residues 389-457) was modelled by AlphaFold2[29]. Full-length TOPOVIBL AlphaFold2-modelled structure (AF-J3QMY9) was then superimposed on the transducer domains of the SPO11-TOPOVIBL complex. Crystal structure of the REC114 PH domain bound the C-terminus of TOPOVIBL determined in this study is shown linked to the SPO11-TOPOVIBL complex via a long-disordered linker. No contact between REC114 PH-TOPOVIBL C-ter and the SPO11-TOPOVIBL complex could be modelled. The modelled complex structure coloured according to the AlphaFold2 per-residue estimate of confidence (pLDDT) and predicted aligned error plots are shown in Supplementary Fig. 16. **b** The AlphaFold2 model (AF-J3QMY9) suggests that the putative ATP-binding site of TOPOVIBL is degenerated. Secondary structure elements forming the ATP-binding site in *S. shibatae* (PDB − 1MX0, lower panel), are predicted to be organised differently in mouse TOPOVIBL.

In particular, the "ATP-lid" possessing key residues ATP binding residues is missing and is replaced by a helical insertion in the transducer domain. In consequence, a corresponding ATP binding site is not formed, and is area is filled with aromatic and hydrophobic residues of TOPOVIBL. TOPOVIBL also lacks the N-terminal "strap" region and the H2TH domain, both involved in the TopoVIB dimerization, indicating that TOPOVIBL dimerization, if occurs, should differ from TopoVIB[39]. **c** The modelled SPO11-TOPOVIBL interface is similar to that of the archaeal TopoVI complex (PDB − 2Q2E, lower panel), with the short additional helix of the TOPOVIBL transducer domain[7]. **d** The modelled SPO11 dimerization interface resembles the one described for archaeal TopoVIA, including the formation of a pseudo-continuous β-sheet formed by the two protomers[41]. **e** The modelled SPO11 catalytic site is equivalent to the one previously described for TopoVIA (PDB − 1D3Y, lower panel) where the two protomers contribute three negatively charged residues that co-ordinate the magnesium atom and the catalytic tyrosine[41].

Such differential effect of the two mutations on the interaction is however not supported by in vitro analysis since no interaction between TOPOVIBL W562A and REC114 PH domain was detected in vitro. It remains possible that in vivo TOPOVIBL W562A can interact weakly with REC114. In vivo interaction assays would be required to test this hypothesis. Alternatively, the stronger phenotype observed in *Top6bl*[Δ17Ct] mice may be due to additional interactions or properties mediated by the last 17 amino acids of TOPOVIBL. The different phenotypes of *Top6bl* mutants, show that REC114 is a regulatory subunit of the activity, and not just an accessory factor of the TOPOVIL complex. The reduced DSB activity in oocytes fits exactly the simple interpretation that REC114 binding to TOPOVIBL is required for the catalytic activity. The male phenotype highlights REC114 double role in fine tuning DSB activity by regulating the timing of DSB formation genome-wide and DSB formation at sub-telomeric regions. It is remarkable that the delayed DSB formation with respect to axis formation in *Top6bl*[Δ17Ct/Δ17Ct] mice does not alter homologous synapsis and seems to allow weak DSB sites to be more active than in wild-type. This could be explained if DSB sites can be turned off upon synapsis, as shown in *S. cerevisiae*[48]. Delaying synapsis would thus allow weak DSB sites to be active. The sub-telomeric effect in *Top6bl*[Δ17Ct/Δ17Ct] mice potentially applies to all chromosomes and is not restricted to the four chromosomes (chr. 4, 9, 13 and X) shown to accumulate REC114, MEI4, IHO1 and ANKRD31 aggregates[27,28]. The property we detected is therefore not strictly linked to those aggregates, which is also consistent with the observation that *Top6bl* mutants do not have any detectable defect in the formation of those aggregates (Supplementary Fig. 11a). We do notice inter-chromosomal quantitative differences of the extent of decreased DSB activity in *Top6bl*[Δ17Ct/Δ17Ct] mice that may at least in part be due to inter-chromosomal differences in hotspot activity within sub-telomeric regions (Supplementary Fig. 13e). Sub-telomeric regions display specific features (i.e., nuclear organization during meiosis[49,50], architecture, organization and other epigenetic properties) that may influence the control of SPO11/TOPOVIBL activity. These regions are known to behave differently from chromosome arms for meiotic DSBs, based on their sensitivity to the expression of a GAL4BD-SPO11 fusion protein on DSB activity[51], to a lower DMC1/SPO11-oligonucleotides ratio[52] and to differential activities in male and female meiosis[53]. It is also possible that mice expressing SPO11β-only (the long isoform of SPO11) and shown to have decreased DSB activity in the PAR, have a specific decrease of DSB activity near autosome ends[38]. This would imply a specific regulation of SPO11 near chromosome ends, as the phenotypes due to the *Top6bl*[Δ17Ct] mutation suggest. As SPO11α (the short isoform of SPO11) is lacking the interaction domain with TOPOVIBL, it is not expected to be catalytically active. SPO11α may rather repress an inhibitor of TOPOVIL. One possibility is that this inhibitor interferes with the REC114-TOPOVIBL interaction. Such scenario, although certainly others can be envisioned, would fit with the potential similarity of phenotypes between *Spo11β*-only and *Top6bl*[Δ17Ct/Δ17Ct] male mice, including the XY synapsis defect. Overall, these observations imply that i) the TOPOVIBL-REC114 interaction is not essential for DSB activity in all genomic contexts; ii) REC114 activity senses or responds to specific chromosomal features. Several studies have highlighted differences of recombination and/or chromosome organization between sexes as well along chromosomes but which links to DSB activity remain to be determined[53,54].

Programming hundreds of DSBs in the genome is a challenge for the cell, and the current knowledge that the catalytic complex of SPO11/TOPOVIBL requires several other proteins is coherent with the need to regulate these events. Here, we described the central role of REC114 in SPO11/TOPOVIBL activity through its direct interaction with TOPOVIBL, thus highlighting a first level of this regulation. Moreover, the DSB program must be executed in two very different cell types

(oocytes and spermatocytes), and our findings show that the REC114-TOPOVIBL interaction is sensitive to these differences. Other components of the meiotic DSB machinery and their potential multiple interactions should contribute to SPO11/TOPOVIBL regulation. Additional directed-mutagenesis studies will unravel them and will identify the complex(es) active in vivo.

## Methods

### Mouse strains
Mice were in the C57BL/6 J background. Mice carrying the homozygous mutant alleles Top6bl < em1(W562A)BdM> and Top6bl < em2(delta17)BdM> were named *Top6bl*[W562A/W562A] and *Top6bl*[Δ17Ct/Δ17Ct], respectively. *Top6bl*[−/−] mice carry the Gm960[em2Arte] allele, a null allele due to a 5 bp deletion in *Top6bl*[5]. Mice housing conditions were: temperature 22 °C, humidity 55%, dark/light cycle 12/12 corresponding to 8 am/8 pm in summer and 7 am/7 pm in winter. All experiments were carried out according to the CNRS guidelines and were approved by the ethics committee on live animals (Comité National de Réflexion Ethique sur l'Expérimentation Animale; project CE-LR-0812 and 1295).

### Generation of mutant mice by CRISPR/Cas9
Mutant mice were created at the Jackson Laboratory using the CRISPR-Cas9 technology with three different guides and two different donor oligos (Supplementary Table 3). Guides were selected to minimize off-target effects. The donor oligos were designed to change the W562 codon TGG (W) to GCG (A), and to introduce a silent mutation (A to G) to generate a *Pst*I restriction site. The Top6bl < em1(W562A)BdM> allele, named *Top6bl*[W562A], is the result of homologous recombination. The Top6bl < em2(delta17)BdM> allele, named *Top6bl*[Δ17Ct], is the result of non-homologous repair and has a 4 bp deletion (Supplementary Fig. 6a). Founders were backcrossed with C57BL/6 J animals to obtain heterozygous animals. The predicted TOPOVIBL protein expressed from each mutant allele and the genotyping strategies are shown in Supplementary Fig. 6.

### Yeast two-hybrid assays: clones, assays, western blotting
All plasmids used in yeast two-hybrid assays were cloned with the Gateway® Gene Cloning Technology (Invitrogen) and transformed in the AH109 and Y187 haploid yeast strains (Clontech). AH109 and Y187 cells were transformed with Gal4 DNA binding domain (GBD) fusion plasmids derived from pAS2 and Gal4 activation domain (GAD) fusion plasmids that were obtained from pGAD. Purified colonies of diploid strains were streaked on synthetic medium (SD) plates lacking leucine and tryptophan (-LW), or leucine, tryptophan and histidine (-LWH), or leucine, tryptophan and histidine with 5 mM amino-triazole (-LWH + 3AT), or leucine, tryptophan, histidine and adenine (-LWHA). Dilution assays were performed by spotting cells on -LW, -LWH, -LWH + 3AT and -LWHA plates that were incubated at 30 °C for 3 days. For verification of protein expression, protein extracts were prepared and analysed by western blotting, as previously described[5], with anti-GAD (1:3000; UPSTATE-06-283) and anti-GBD (1:1000; SIGMA; G3042) antibodies.

### Protein expression, purification and crystallization
His-tagged mouse REC114 15-159 was expressed in *E. coli* BL21-Gold (DE3) cells (Agilent) from the pProEXHTb expression vector (Invitrogen). The protein was first purified by affinity chromatography using the Ni[2+] resin (Chelating Sepharose, GE Healthcare). After His-tag cleavage with the TEV protease, it was purified through a second Ni[2+] column and size-exclusion chromatography on Superdex 200 (GE Healthcare). The pure protein was concentrated to 20 mg ml[−1] in a buffer containing 20 mM Tris, pH 8.0, 200 mM NaCl and 5 mM β-mercaptoethanol, and supplemented with a three-fold molar excess of TOPOVIBL peptide (559-EDLWLQEVSNLSEWLNPG-576). The complex

was crystallized using the hanging drop vapor diffusion method at 20 °C. The best diffracting crystals grew within seven days in a solution containing 1.6 M $MgSO_4$, 100 mM MES (pH 6.5), and 10% (v/v) dioxane. For data collection at 100 K, crystals were snap-frozen in liquid nitrogen with a solution containing mother liquor and 25% (vol/vol) glycerol.

## Data collection and structure determination

Crystals of the mouse REC114-TOPOVIBL complex belong to the space group $P6_122$ with the unit cell dimensions $a$, $b$ = 108.7 Å and $c$ = 83. Å. The asymmetric unit contains one REC114-TOPOVIBL dimer and has a solvent content of 68%. A complete native dataset was collected to a 2.5 Å resolution, partially extending to 2.26 Å on the ESRF beamline ID30B using the MXcuBE3 software (ESRF). The data were processed using autoPROC[55]. Phases were obtained by molecular replacement using PHASER[56] with the crystal structure of the REC114 PH domain (PDB code: 6HFG) as search model. The initial map was improved using the prime-and-switch density modification option of RESOLVE[57]. After manual model rebuilding with COOT[58], the structure was refined using Refmac5[59] to a final $R$-factor of 22.6% and $R_{free}$ of 24.9% (Supplementary Table 1) with all residues in the allowed (95.2% in favored) regions of the Ramachandran plot, as analysed by MOLPROBITY[60]. A representative part of the $2F_o - F_c$ electron density map covering the TOPOVIBL-REC114 interface is shown in Supplementary Fig. 2b.

## Pull-down assays

Full-length REC114 and its variants were cloned as Strep-tag fusions (using a single Strep-tag: WSHPQFEK) into pRSFDuet-1 (Novagen). TOPOVIBL[1–385] was cloned as a 6xHis-SUMO fusion protein in pETM11, and TOPOVIBL[452–579] and its mutated versions as 6xHis fusion proteins in pProEXHTb. Proteins were expressed individually in *E. coli* BL21Gold (DE3) cells. All proteins were first purified by affinity chromatography (Strep-Tactin XT resin (IBA), Ni-Chelating Sepharose, GE Healthcare) and gel filtration on Superdex 200 (GE Healthcare). Mixtures of REC114 and TOPOVIBL were loaded onto Strep-Tactin XT (IBA) resin columns. Columns were then extensively washed with a buffer containing 100 mM Tris pH 8, 100 mM NaCl, 5 mM β-mercaptoethanol. Bound proteins were eluted by addition of 50 mM of D-Biotin, and analysed on SDS-PAGE. ANKRD31[1808–1857] was cloned as 6xHis-MBP fusion in pETM41. Strep-REC114[1–159] and ANKRD31[1808–1857] were individually expressed in *E. coli* BL21Gold (DE3) cells. Following cell disruption, supernatants containing soluble Strep-REC114[1–159] and ANKRD31[1808–1857] were mixed. The complex was further purified by Superdex 200 (GE Healthcare) gel filtration column. The pull-down with increasing amounts of TOPOVIBL[452–579] on Strep-Tactin XT (IBA) resin column was performed as above. For pull-down presented in Fig. 2g, a total of 0.8% of the input (lanes 1–6) and 1.2% of the eluates (lanes 7–14) were analyzed on 12% SDS-PAGE gels stained with coomassie brilliant blue. Control lanes 7 and 14 show REC114 alone. For pull-down presented in Fig. 2k, a total of 0.3% of the input (lanes 1–6) and 1.2% of the eluates (lanes 7–14) were analyzed on 12% SDS-PAGE gels stained with coomassie brilliant blue.

## Isothermal Titration Calorimetry (ITC)

ITC experiments were performed at 25 °C using an ITC200 microcalorimeter (MicroCal). Experiments included one 0.5 µl injection and 18-20 injections of 1.5-2 µL of 0.3-1.8 mM TOPOVIBL (TOPOVIBL[452–579], TOPOVIBL[452–562,W562A,563–579], TOPOVIBL[559–576] or TOPOVIBL[559–562,W562A,563–576]) into the sample cell that contained 30-40 µM REC114[15–159] or Strep-REC114[1–159] in 20 mM Tris (pH 8.0), 100 mM NaCl, 5% glycerol and 5 mM β-mercaptoethanol. The initial data point was deleted from the data sets. Binding isotherms were fitted with a one-site binding model by nonlinear regression using the Origin software, version 7.0 (MicroCal).

## Preparation of mouse protein extracts, immunoprecipitation and western blotting

Whole cell protein extracts were prepared from eight frozen testes collected at 14 dpp for each genotype. After protein extraction by homogenizing cells with a Dounce homogenizer in HNTG buffer (150 mM NaCl, 20 mM HEPES pH7.5, 1% Triton X100, 10% glycerol, 1 mM MgCl, Complete protease Inhibitor (Roche 11873580001)), followed by sonication, benzonase (250 U) was added at 4 °C for 1 h. After centrifugation (16000 g, 4 °C, 10 min) to remove debris, immunoprecipitation was performed with 5 µg of homemade anti-TOPOVIBL antibody. For each immunoprecipitation, 3.5 mg of whole cell protein extract and 50 µl of Protein A Dynabeads (Invitrogen 10001D) were used. Then, immunoprecipitates were resuspended in 40 µl of Laemmli buffer and TOPOVIBL immunoprecipitation was assessed by western blotting with a homemade affinity-purified anti-TOPOVIBL (1/1000) antibody followed by an anti-rabbit LC mouse monoclonal secondary antibody (Jackson ImmunoResearch 211-032-171, 1/3000).

## RT-PCR assays

Total RNA was extracted with the miRNeasy Mini Kit (Qiagen) according to the manufacturer's instructions. For RT-PCR, first-strand DNA was synthesized using oligo d(T)[18] (Ambion), SuperScriptIII (Invitrogen), and total RNA (1-2 µg) from 16dpc ovaries.

The open reading frames of *Top6bl* and *Spo11* were amplified using standard PCR conditions and the primer pairs Oli63/Oli70 and Spo11:116U22/Spo11:655L22, respectively (Supplementary Table 3). PCR cycling conditions were: 3 min at 94 °C, 35 cycles of 30 sec at 94 °C, 30 sec at 54 °C, and 2 min or 30 sec at 72 °C, followed by 5 min at 72 °C. *Top6bl* ORF was then digested with the *Eci*I enzyme.

## Histological analysis of paraffin sections and TUNEL assay

Mouse testes or ovaries were fixed in Bouin's solution for Periodic Acid Schiff (PAS) staining of testes and haematoxylin eosin staining of ovaries. Fixation was in 4% paraformaldehyde/1X PBS for immunostaining and TUNEL assay. Testes and ovaries were embedded in paraffin and cut in 3µm-thick sections. Sections were scanned using the automated tissue slide-scanning tool of a Hamamatsu NanoZoomer Digital Pathology system. TUNEL assay was performed with the DeadEnd Fluorometric TUNEL System (Promega), according to the manufacturer's protocol.

## Spermatozoid counting

After dissection of the epididymis caudal part from adult testes (2 month-old), spermatozoids were extracted from the epididymis by smashing or crushing the tissue in PBS. After homogenization by pipetting, 10 µl of the soluble part was diluted in 1 mL of water, and spermatozoids were counted.

## Immunocytology

Spread from spermatocytes and oocytes were prepared with the dry down technique, as described[61]: Briefly, a suspension of testis cells was prepared in PBS, and then incubated in a hypotonic solution for 8 min at room temperature. Cells were centrifuged, resuspended in a solution of 66 mM sucrose and spread on slides with 1% paraformaldehyde, 0.05% Triton. Nuclei were dried for 1 to 2 h in a humid chamber. Immunostaining was performed using a milk-based blocking buffer (5% milk, 5% donkey serum in PBS)[62]. Primary antibodies were incubated overnight at room temperature. Secondary antibodies were incubated at 37 °C for 1.5 h. Nuclei were stained with DAPI (4'−6-Diamidino-2-phenylindole, 2 µg/ml) during the final washing step.

## Antibodies

Guinea pig anti-SYCP3[62] (1/500), rabbit anti-SYCP1 (Abcam, 15090, 1/400), rabbit anti-DMC1 (Santa Cruz, H100, 1/200) anti-RPA2 (Abcam, ab76420 clone name EPR2877Y, 1/200), anti-MEI4[23] (1/100),

anti-REC114[25] (1/50), anti-REC114 (gift from S. Keeney, 1/1000), anti-ANKRD31[28] (1/400), anti-IHO1[26] (1/2000) and mouse monoclonal anti-phospho-histone H2AX (Ser139) (γH2AX) (Millipore, 05-636, 1/10000) antibodies were used for immunostaining. Homemade affinity purified anti-TOPOVIBL antibody: rabbits were injected with full-length mouse His-TOPOVIBL protein prepared from *E. coli* inclusion bodies. Rabbit serum was purified by affinity using His-TOPOVIBL purified from inclusion bodies.

## Metaphase spread preparation

Tubules from decapsulated testes were pulled apart in 1% trisodium citrate and transferred into a 15 ml tube. After pipette homogenization and 1 min sedimentation, the cell-containing supernatant was transferred in a new 15 ml tube. Following the same procedure, the tubule pieces were rinsed twice with 3 ml of 1% trisodium citrate. The cell solution was centrifuged at 180 g for 10 min, and the pellet resuspended in 100 µl of supernatant. Then, 3 ml of methanol: acetic acid: chloroform (3:1:0.05) solution was added drop by drop to the cell solution (rolling the first drops down the sides of the tube while flicking the tube). Cells were then centrifuged at 180 g for 10 min and resuspended in 100 µl of supernatant, and 3 ml of methanol: acetic acid (3:1) was added to the tube. After 10 min of incubation at room temperature, cells were centrifuged again (180 g for 10 min) and resuspended in ~1 ml methanol: acetic acid (3:1). To prepare the slides, 40 µl of the cell suspension was dropped from a height of ~40 cm onto a slide that was held titled at 45°. Slides were dried in a humid chamber.

## FISH for chromosome painting

X (D-1420-050-FI; D-1420-050-OR) and Y (D-1421-050-FI; D-1421-050-OR) chromosome-specific probes were used according to the manufacturer (Metasystems Probes). 10 µl of probe mixture were added onto slides with metaphase spreads, covered with coverslips and sealed. The samples were then denatured, hybridized, washed and stained with DAPI as recommended by the manufacturer (Metasystems Probes).

## Image analysis

For focus quantification, all images were deconvoluted using the Huygens software. Image J was used to quantify foci that colocalized with the chromosome axis defined by SYCP3 staining.

For γH2AX quantification in oocytes, signal intensity was obtained using Cell Profiler on non-deconvoluted images. Integrated intensity was used for the analysis. For spermatocytes, both Cell Profiler and Image J quantifications were performed and gave similar results. The output of Image J integrated intensity is presented.

Staging criteria were as follows. Pre-leptotene nuclei had weak SYCP3 nuclear signal and no or very weak γH2AX signal; early leptotene nuclei were γH2AX-positive and with only short SYCP3 fragments; mid leptotene nuclei were γH2AX-positive and with short and long SYCP3 fragments; late leptotene were γH2AX-positive and with only long SYCP3 fragments; zygotene nuclei had partially synapsed homologs; and pachytene cells had all 19 autosomes fully synapsed (Supplementary Fig. 10a).

## DMC1-SSDS analysis

**Library preparation and sequencing.** DMC1 ChIP-seq was performed as described in[63] using a goat anti-DMC1 antibody (0.5 mg/ml; Santa Cruz, C-20). Six *Top6bl*$^{\Delta 17Ct/\Delta 17Ct}$ testes and two wild-type testes from 12 to 25-week-old mice were used for each replicate. Sequencing was performed on a HiSeq 2500 instrument in paired-end mode (2x150bp).

**DMC1-SSDS mapping and hotspot identification.** After quality control and read trimming to remove adapter sequences and low-quality reads, DMC1 ChIP-SSDS reads were mapped to the UCSC mouse genome assembly build GRCm38/mm10. The previously published method[35] was used for DMC1-SSDS read mapping (i.e., the BWA modified algorithm and a customized script that were specifically developed to align and recover ssDNA fragments). A filtering step was performed on the aligned reads to keep only non-duplicated and high-quality uniquely mapped reads with no more than one mismatch per read. To identify meiotic hotspots from biologically replicated samples in DMC1-SSDS, the Irreproducible Discovery Rate (IDR) method was used, as done in our previous studies. This method was developed for ChIP-seq analysis and extensively used by the ENCODE and modENCODE projects[64]. The framework developed by Qunhua Li and Peter Bickel's group (https://sites.google.com/site/anshulkundaje/projects/idr) was followed. Briefly, this method allows testing the reproducibility within and between replicates by using IDR statistics. Following their pipeline, peak calling was performed using MACS version 2.0.10 with relaxed conditions (−pvalue = 0.1−bw1000−nomodel−shift400) for each of the two replicates, the pooled dataset, and pseudo-replicates that were artificially generated by randomly sampling half of the reads twice for each replicate and the pooled dataset. Then IDR analyses were performed, and reproducibility was checked. Final peak sets were built by selecting the top N peaks from pooled datasets (ranked by increasing p values), with N defined as the highest value between N1 (the number of overlapping peaks with an IDR below 0.01, when comparing pseudo-replicates from pooled datasets) and N2 (the number of overlapping peaks with an IDR below 0.05 when comparing the true replicates, as recommended for the mouse genome). Hotspot centring and strength calculation were computed following the method described by Khil et al.[35]. All read distributions and signal intensities presented in this work were calculated after pooling reads from both replicates, if not otherwise stated. When DSB maps were compared between mouse genotypes, the 1bp-overlaps were restricted to the central 400 bp of hotspots (+/− 200 bp around the peak centre). For correlation plots, the type 1 single-strand DNA signal was library-normalized (fragment per million).

**Differential analysis of hotspot strength.** To compare hotspot usage between *Top6bl*$^{\Delta 17Ct/\Delta 17Ct}$ and wild-type mice, DMC1-SSDS signal intensity was compared at the 13562 wild-type hotspots using DESeq2[65]. Among these hotspots, 4207 hotspots (31%) were filtered out with the default independent filtering option using the mean of normalized counts as filter statistic. The aim was to remove sites with too low counts (mean count below 22) that have zero or low chance of showing significant differences to increase the detection power for the other sites. For the 9355 tested hotspots, log fold change shrinkage was performed to correct data dispersion using the *apeglm* method[66]. p-value were adjusted for multiple testing within DESeq2 using the procedure of Benjamini and Hochberg. The hotspots with increased or decreased DMC1-SSDS signal intensity were then determined using an adjusted p-value threshold of 0.1 and a log2 fold change value below or above zero, respectively. This led to the identification of 214 increased (1.2% of total hotspots, and 2% of the tested ones), 1098 decreased (8% of total hotspots, and 12% of the tested ones), and 8043 unchanged hotspots in *Top6bl*$^{\Delta 17Ct/\Delta 17Ct}$ mice (Fig. 6c).

**Analysis of hotspot distribution at sub-telomeric regions.** Visual inspection of DMC1-SSDS signal intensity along chromosomes and the localization of hotspots with decreased DMC1-SSDS signal intensity in *Top6bl*$^{\Delta 17Ct/\Delta 17Ct}$ suggested that much of the DMC1 signal decrease was located near the q-arm telomeres (telomeres distant from centromeres). Note that as the genomic DNA sequence of p-arms has not been assembled, it is immediately flanked by centromeric and q-arm DNA sequences, where the signal can be mapped and quantified. To test whether this biased distribution was significant, each hotspot was annotated as sub-telomeric when within the sub-telomeric region defined with a variable size from 1 to 10 Mb. For each sub-telomeric region, the numbers of unchanged, decreased and increased hotspots

in the sub-telomeric versus the non-sub-telomeric region (i.e., the rest of the chromosome) were counted. Pearson's Chi-square tests were computed (by taking into account or not the increased hotspots) and p-values were adjusted for multiple testing using the Benjamini & Yekutieli method. Megabase-normalized counts (hotspot density) were measured and plotted (Fig. 6g for sub-telomeric regions of 3 Mb). For each chromosome, the sub-telomeric over non-sub-telomeric ratio of hotspot density was calculated for each unchanged, decreased or increased hotspot category (Supplementary Fig. 13d). Alternatively, to evaluate hotspot activity without a fixed distance from the telomere, for each chromosome, the $Top6bl^{\Delta17Ct/\Delta17Ct}$/wild-type signal ratio was averaged over the ten most q-arm telomeric hotspots, and then compared to the averaged signal ratio measured over another set of ten consecutive hotspots randomly chosen along the chromosome (excluding the last tens). Then, the ratio of these two mean values was computed. The procedure was repeated 10 times, each time with a different random set of 10 non-sub-telomeric hotspots. The sub-telomeric effects are presented in Fig. 6f as the distribution of these ratios. As control, these ratios were computed not between the last ten and ten non-sub-telomeric hotspots, but between two non-sub-telomeric random hotspot sets (procedure repeated 10 times) (Supplementary Fig. 13c).

### Statistical analysis

The statistical analyses of cytological observations were done with GraphPad Prism 9. The nonparametric Mann-Whitney test was used to compare the number of foci, and the Pearson's Chi square test to compare distributions, as indicated in the figure legends. The Chi square tests were performed at http://vassarstats.net/ and https://www.quantitativeskills.com/sisa/index.htm. Box plots (25–75 percentiles) show the median and 5-95 percentiles. Statistical tests for DMC1-SSDS data were done using R version 4.0.3. All tests and $p$ values (n.s., not significant. $*P < 0.05$, $**P < 0.01$, $***P < 0.001$, $****P < 0.0001$) are provided in the corresponding legends and/or figures.

### Statistics and reproducibility

The pull-down assays shown in Figs. 1d, 2g, were performed twice. The pull-down assays shown Fig. 2k and Supplementary Fig. 4b were performed three times. The gel filtrations shown in Supplementary Figs. 2a, 4d, 5b, c, 6d were performed at least twice. IP from Supplementary Fig. 6e was reproduced three times. RT-PCR from Supplementary Fig. 6f were performed twice per genotype.

### Reporting summary

Further information on research design is available in the Nature Portfolio Reporting Summary linked to this article.

## Data availability

The DMC1-SSDS raw and processed data for this study have been deposited in the European Nucleotide Archive (ENA) at EMBL-EBI and are available through the project identifier PRJEB43730. The atomic coordinates and structure factors of the mouse REC114-TOPOVIBL complex determined in this study have been deposited at the Protein Data Bank (http://www.ebi.ac.uk/pdbe/) under the PDB accession code 7QWV. Additional information, resources and reagents are available from the corresponding author upon reasonable request. Source data are provided with this paper.

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

## Acknowledgements

We thank the following Biocampus facilities from Montpellier for their service: the Réseau des Animaleries de Montpellier (RAM) for animal care, the management of our mouse strains by Manon Leportier, the Réseau d'Histologie Expérimentale de Montpellier (RHEM) for histology,

and Montpellier Resources Imagerie (MRI) for microscopy. We thank all lab members for discussion and Pauline Auffret for support in bioinformatic analysis. We thank Scott Keeney for the anti-REC114 antibody, Attila Toth for the anti-ANKRD31 and anti-IHO1 antibodies. We thank Caroline Mas for assistance and access to the biophysics platform, the staff of the ESRF-EMBL (European Synchrotron Radiation Facility-European Molecular Biology Laboratory) Joint Structural Biology Group, particularly Andrew McCarthy and Matthew Bowler, for access to and help with the ESRF beamlines. We thank the EMBL high-throughput crystallization facility (HTX). BdM and JK were funded by ANR Topo-breaks (ANR- 18-CE11-0024-01). BdM was also funded by Prize Coups d'Élan for French Research from the Fondation Bettencourt-Schueller, ERC (European Research Council (ERC) Executive Agency under the European Community's Seventh Framework Programme (FP7/2007-2013 Grant Agreement no. 322788) and under the European Union's Horizon 2020 research and innovation programme (Grant Agreement no. 883605)) and CNRS. We acknowledge also the CNRS-Plan Cancer ATIP-Avenir program funding to JK, and the CNRS INSERM ATIP-Avenir 2017 program funding to TR. Ariadna B. Juarez-Martinez was supported by the Labex GRAL (Grenoble Alliance for Integrated Structural Cell Biology) (ANR-10-LABX-49-01) and the People Program (Marie Curie Actions) of the European Union's Seventh Framework Program (FP7/2007–2013) under REA grant agreement PCOFUND-GA-2013-609102, through the PRESTIGE program coordinated by Campus France. Financial support from the Center National de la Recherche Scientifique (IR-RMN-THC Fr3050) is gratefully acknowledged. IBS acknowledges integration into the Interdisciplinary Research Institute of Grenoble (IRIG, CEA). This work used the platforms of the Grenoble Instruct-ERIC centre (ISBG; UAR 3518 CNRS-CEA-UGA-EMBL) within the Grenoble Partnership for Structural Biology (PSB), supported by FRISBI (ANR-10-INBS-0005-02) and GRAL, financed within the University Grenoble Alpes graduate school (Ecoles Universitaires de Recherche) CBH-EUR-GS (ANR-17-EURE-0003).

## Author contributions

J.K., T.R. and B.dM. designed the project. Acquisition and analysis of the structural and biochemical studies were by A.J.-M. and J.K. with some complementary experiments by H.L. A.N. performed most Y2H and mouse experiments. T.R. and B.dM. performed some image acquisition and analysis and interpreted the data. C.B. performed immunocytochemistry. B.D. performed immunoprecipitations experiments. C.G. performed DMC1 Chip-Seq. DMC1 SSDS was analyzed by J.C. H.M.B. analyzed the conservation of TOPOVIBL and identified the conserved motif. H.M.B. and J.K. generated the Alfafold2 models. B.dM. wrote the first draft of the manuscript; B.dM., T.R. and J.K. revised the manuscript.

## Competing interests

The authors declare no competing interests.
