## [Peer Review File · Nature Communications]

TOPOVIBL-REC114 interaction regulates meiotic DNA double-strand breaksEditorial Note: This manuscript has been previously reviewed at another journal that is not operating a transparent peer review scheme. This document only contains reviewer comments and rebuttal letters for versions considered at *Nature Communications* .

REVIEWERS' COMMENTS

Reviewer #1 (Remarks to the Author):

The authors have done a good job responding to the very detailed comments of all three reviewers. I'm now satisfied that acceptance and publication can proceed. Congratulations to the authors on a fine study.

Reviewer #2 (Remarks to the Author):

The authors have addressed all of my previous points; the paper is excellent. I stated before, is an important breakthrough since we now have molecular detail of the TOPOVIL complex interactions with the regulatory subunits. Congratulations!

Reviewer #3 (Remarks to the Author):

In this article, the authors uncovered an interaction between two proteins required for meiotic recombination initiation via programmed DSB formation, TOPOVIBL and REC114, whose precise contribution is elusive. By crystallizing the interacting peptide of TOPOVIBL with a REC114 domain whose structure was determined in a previous study, they identify sites of interaction on each partner. They generated separation of function mutant mice affecting this interaction and uncovered a meiotic recombination phenotype, differing between male and female meiosis. Although meiotic DSB are severely reduced in females, which are sterile or subfertile, in males, DSBs are only slightly redistributed, lowered close to telomeres, and males are subfertile. Finally, the TOPOVIBL protein has not been crystallized, but using alpha-fold for modeling its structure together with the other, SPO11, subunit, the authors predict that it does not likely have an ATPase activity, differently from its remote homolog TopoVIB subunit. This adds some mechanistic information about the process of meiotic DSB catalysis.

In this revised manuscript, the authors addressed satisfactorily all my previous major and minor points.

In particular, they more thoroughly describe what was known for the *S. cerevisiae* proteins and how the interactions and the meiotic DSB protein complexes differ during evolution.

Also, they clarify the differences between their two mutants and why they give a different phenotype. They also added more detailed introduction and discussion. As a whole, the paper is a lot improved, and now suitable for publication.